# HollowFlow: Efficient Sample Likelihood Evaluation using Hollow Message Passing

**Johann Flemming Gloy**[1]     **Simon Olsson**[1*]
[1]Department of Computer Science and Engineering,
Chalmers University of Technology and University of Gothenburg
SE-41296 Gothenburg, Sweden.
{gloy, simonols}@chalmers.se

## Abstract

Flow and diffusion-based models have emerged as powerful tools for scientific applications, particularly for sampling non-normalized probability distributions, as exemplified by Boltzmann Generators (BGs). A critical challenge in deploying these models is their reliance on sample likelihood computations, which scale prohibitively with system size $n$, often rendering them infeasible for large-scale problems. To address this, we introduce *HollowFlow*, a flow-based generative model leveraging a novel non-backtracking graph neural network (NoBGNN). By enforcing a block-diagonal Jacobian structure, HollowFlow likelihoods are evaluated with a constant number of backward passes in $n$, yielding speed-ups of up to $\mathcal{O}(n^2)$: a significant step towards scaling BGs to larger systems. Crucially, our framework generalizes: **any equivariant GNN or attention-based architecture** can be adapted into a NoBGNN. We validate HollowFlow by training BGs on two different systems of increasing size. For both systems, the sampling and likelihood evaluation time decreases dramatically, following our theoretical scaling laws. For the larger system we obtain a $10^2\times$ speed-up, clearly illustrating the potential of HollowFlow-based approaches for high-dimensional scientific problems previously hindered by computational bottlenecks.

## 1   Introduction

Efficiently sampling high-dimensional, non-normalized densities — a cornerstone of Bayesian inference and molecular dynamics — remains computationally intractable for many scientific applications. Boltzmann generators (BGs) [1] address this by training a surrogate model $\rho_1(\mathbf{x})$ to approximate the Boltzmann distribution $\mu(\mathbf{x}) = Z^{-1}\exp(-\beta u(\mathbf{x}))$, where $u(\mathbf{x})$ is the potential energy of the system configuration $\mathbf{x} \in \Omega \subset \mathbb{R}^N$, $\beta$ the inverse temperature, and $Z$ the intractable partition function. While BGs enable unbiased estimation of observables through reweighting $\mathbf{x}_i \sim \rho_1$ with weights $w_i \propto \mu(\mathbf{x}_i)/\rho_1(\mathbf{x}_i)$, their practicality is limited by a fundamental trade-off: likelihood computations of expressive surrogates rely on $N$ automatic differentiation backward passes with system dimensionality $N$, rendering them infeasible for large-scale problems.

To address this problem, we outline a framework combining non-backtracking equivariant graph neural networks with continuous normalizing flows (CNFs), breaking this trade-off. This strategy enforces a Jacobian that can easily be split into a **block-diagonal** and a **block-hollow** part, reducing the number of backward passes required for likelihood computation to scale as $\mathcal{O}(1)$ in $N$. Our contributions include:

>**Hollow Message Passing (HoMP):** A general scheme for message passing with a block-diagonal Jacobian structure.

---

[*]Corresponding author

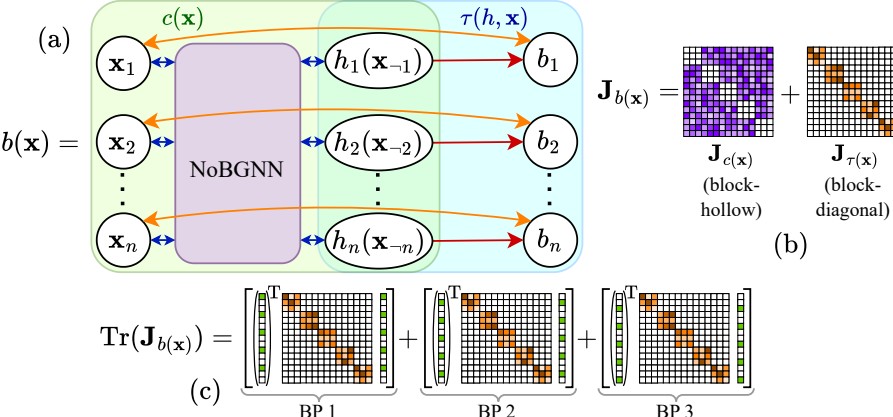

Figure 1: **Summary of HollowFlow**: (a) A vector field $b$ is parametrized with an (equivariant) non-backtracking graph neural network (NoBGNN) and a series of (equivariant) neural networks $\tau_i$, such that its Jacobian $\mathbf{J}_b$ (b) can be decomposed into a block-hollow and block-diagonal part with block size $d$ (in this example, $d = 3, n = 5$). (c) Efficient evaluation of the trace of the Jacobian with only $d$ instead of $nd$ backward passes (BP) through the network. The vector components are one where the entry is green and zero elsewhere.

**Theoretical Proofs and Guarantees:** A proof of the block-diagonal structure of HoMP and scaling laws of the computational complexity of forward and backward passes.

**Scalable Boltzmann Generators:** A HollowFlow-based BG implementation achieving $10^1$–$10^2\times$ faster sampling and likelihood evaluation, e.g., $10^2\times$ for LJ55, a system of 55 particles in 3-dimensions interacting via pair-wise Lennard-Jones potentials.

Experiments on multi-particle systems validate HollowFlow's scalability, bridging a critical gap in high-dimensional generative modelling for the sciences.

## 2 Background

### 2.1 Boltzmann Generators and Observables

Computing observables by averaging over the Boltzmann distribution is a central challenge in chemistry, physics, and biology. The Boltzmann distribution is given by,

$$\mu(\mathbf{x}) = Z^{-1} \exp(-\beta u(\mathbf{x})) \tag{1}$$

where $\beta = (k_\mathrm{B}T)^{-1}$ is the inverse thermal energy and $u : \Omega \to \mathbb{R}$ is the potential energy of a system configuration $\mathbf{x} \in \Omega \subseteq \mathbb{R}^N$.

Observables are experimentally measurable quantities, such as free energies [2–4]. Generating ensembles which align with certain observables can give us fundamental insights into the physics of molecules, including mechanisms, rates, and affinities of drugs binding their targets [5, 6], protein folding [7, 8], or phase transitions [9–12].

The primary strategies for generating samples from the Boltzmann distribution involve Markov Chain Monte Carlo (MCMC) or molecular dynamics (MD) simulations. These approaches generate unbiased samples from the Boltzmann distribution asymptotically, however, mix slowly on the complex free energy landscape of physical systems, where multiple, metastable states are separated by low-probability channels. In turn, extremely long simulations are needed to generate independent sampling statistics using these methods. In the molecular simulation community, this problem has stimulated the development of a host of enhanced sampling methods, for a recent survey, see [13].

Boltzmann Generators [1] instead try to learn an efficient surrogate from biased simulation data, and recover unbiased sampling statistics through importance weighting. The key advantage of BGs is that sampling using modern generative methods, such as normalizing flows, is much faster than the conventional simulation based approaches, which would enable amortization of the compute cost per Boltzmann distributed sample.

## 2.2 Normalizing Flows

To enable reweighting, evaluating the sample likelihood under a model, $\rho_1$, is critical. A *normalizing flow* (NF) [14, 15] allows efficient sampling, *and* sample likelihood evaluation, by parameterizing a diffeomorphism, $\phi_\theta : \mathbb{R}^N \to \mathbb{R}^N$, with a constrained Jacobian structure. The diffeomorphism (or *flow*) $\phi_\theta$ maps an easy-to-sample base distribution, $\rho_0$ to an approximation $\rho_1$ of the empirical data density, $p_D$, e.g., by minimizing the forward Kullback-Leibler divergence, $D_{\mathrm{KL}}(p_D||\rho_1) = \mathbb{E}_{\mathbf{x} \sim p_D}[\log p_D(\mathbf{x}) - \log \rho_1(\mathbf{x})]$. The change in log-density when transforming samples with $\phi_\theta$ is given by

$$\Delta \log \rho^{\mathrm{NF}}(\mathbf{x}) := \log \rho_1(\mathbf{x}) - \log \rho_0(\phi_\theta^{-1}(\mathbf{x})) = \log \left| \det \frac{\partial \phi_\theta^{-1}(\mathbf{x})}{\partial \mathbf{x}} \right|. \tag{2}$$

Computing the log-determinant of a general $N \times N$ Jacobian is computationally expensive ($\mathcal{O}(N^3)$), however, endowing the Jacobian $\frac{\partial \phi_\theta^{-1}(\mathbf{x})}{\partial \mathbf{x}}$ with structure, i.e., triangular [16–18], can make the likelihood evaluation fast.

**Continuous Normalizing Flows**   A flow can also be parametrized as the solution to an initial value problem, specified by a velocity field $b_\theta(\mathbf{x}, t) : \mathbb{R}^N \times [0, 1] \to \mathbb{R}^N$ of an ordinary differential equation (ODE)

$$\mathrm{d}\mathbf{x}(t) = b_\theta(\mathbf{x}(t), t) \, \mathrm{d}t, \; \mathbf{x}(0) \sim \rho_0, \; \mathbf{x}(1) \sim \rho_1. \tag{3}$$

The general solution to this problem is a flow, $\phi_\theta(\mathbf{x}(t), t) : \mathbb{R}^N \times [0, 1] \to \mathbb{R}^N$, which satisfies the initial condition, e.g., $\phi_\theta(\mathbf{x}(0), 0) = \mathbf{x}(0)$. Such a model is known as a *continuous normalizing flow* (CNF). The ODE eq. (3) together with the prior $\rho_0$ gives rise to a time-dependent distribution $\rho(\mathbf{x}(t), t)$ given by the continuity equation

$$\partial_t \rho_t = -\nabla \cdot (\rho_t b_\theta(\mathbf{x}, t)), \tag{4}$$

with $\partial_t$ and $\nabla \cdot$ denoting the partial derivative w.r.t. $t$ and the divergence operator, respectively. Solving eq. (4) we get the change in log-probability for the CNF,

$$\Delta \log \rho^{\mathrm{CNF}} := \log \rho_1(\mathbf{x}(1)) - \log \rho_0(\mathbf{x}(0)) = -\int_0^1 \nabla \cdot b_\theta(\mathbf{x}(t), t) \, \mathrm{d}t. \tag{5}$$

In principle, CNFs allow for much more expressive architectures to be used to parametrize $b_\theta$.

The currently most scalable approach to train CNFs is conditional flow matching (CFM) or related objectives [19–22]. In CFM, the intractable flow matching loss, where $b_\theta$ is regressed directly against the unknown true vector field $u_t(\mathbf{x})$, of a flow $\phi$, is replaced by the conditional flow matching loss which has the same gradient w.r.t. the parameters $\theta$ [20]:

$$\mathcal{L}_{\mathrm{CFM}} = \mathbb{E}_{t \sim \mathcal{U}[0,1], \mathbf{x} \sim p_t(\mathbf{x}|\mathbf{z}), \mathbf{z} \sim p(\mathbf{z})} ||b_\theta(\mathbf{x}, t) - u_t(\mathbf{x}|\mathbf{z})||_2^2. \tag{6}$$

The true vector field $u_t(\mathbf{x})$ and its corresponding probability density $p_t(\mathbf{x})$ are given as marginals over the conditional probability and an arbitrary conditioning distribution $p(\mathbf{z})$:

$$p_t(\mathbf{x}) = \int p_t(\mathbf{x}|z)p(\mathbf{z}) \, \mathrm{d}\mathbf{z} \quad \text{and} \quad u_t(\mathbf{x}) = \int \frac{p_t(\mathbf{x}|\mathbf{z})u_t(\mathbf{x}|\mathbf{z})}{p_t(\mathbf{x})} p(\mathbf{z}) \, \mathrm{d}\mathbf{z}. \tag{7}$$

where we parameterize $p_t(\mathbf{x}|\mathbf{z})$ and $u_t(\mathbf{x}|\mathbf{z})$ with a linear interpolant with connections to optimal transport theory [22]:

$$z = (\mathbf{x}_0, \mathbf{x}_1), \quad p(\mathbf{z}) = \pi(\mathbf{x}_0, \mathbf{x}_1) \quad p_t(\mathbf{x}|\mathbf{z}) = \mathcal{N}(t\mathbf{x}_1 + (1 - t)\mathbf{x}_0, \sigma), \quad u_t(\mathbf{x}|\mathbf{z}) = \mathbf{x}_1 - \mathbf{x}_0, \tag{8}$$

where the coupling $\pi(\mathbf{x}_1, \mathbf{x}_0)$ minimizes the 2-Wasserstein optimal transport map between the prior $\rho_0(\mathbf{x}_0)$ and the target $\mu(\mathbf{x}_1)$ using a mini-batch approximation [20, 22, 23].

Unfortunately, akin to general Jacobian log-determinant calculation of normalizing flows, computation of $\nabla \cdot b_\theta(\mathbf{x}(t), t)$ (eq. (5)) requires $\mathcal{O}(N)$ backward passes, for each numerical integration step along $t$, making the evaluation of sample likelihoods impractical in high-dimensions, and consequently, their use for BGs limited.

**Equivariant flows** Symmetries can be described in terms of a group $\mathcal{G}$ that acts on a finite-dimensional vector space $V \cong \mathbb{R}^d$ through a group representation $\rho : \mathcal{G} \to GL(d, \mathbb{R})$. A function $f : \mathbb{R}^d \to \mathbb{R}^d$ is called $\mathcal{G}$-equivariant if $f(\rho(g)x) = \rho(g)f(x)$ and $h : \mathbb{R}^d \to V'$ is called $\mathcal{G}$-invariant if $h(\rho(g)\mathbf{x}) = h(\mathbf{x})$ for all $g \in \mathcal{G}$ and $\mathbf{x} \in \mathbb{R}^d$ where $V'$ is another vector space. If the vector field $b_\theta$ of a CNF is $\mathcal{H}$-equivariant, where $\mathcal{H} < \mathcal{G}$ and the prior density $\rho_0$ is $\mathcal{G}$-invariant it has been shown that the push forward density, of the flow $\phi$, $\rho_1$ is $\mathcal{H}$-invariant [23–25]. In practical applications, symmetries under the Euclidean group $E(3)$ are of particular interest as these are the symmetries that are naturally obeyed by many physical systems such as molecules. Thus, several GNN architectures have been proposed that are equivariant or invariant under the action of $E(3)$ or some subgroup of $E(3)$ [26, 27].

## 2.3 HollowNets

Reverse-mode automatic differentiation, or back-propagation, allows for the computation of vector-Jacobian products with a cost which is asymptotically equal to that of the forward pass [28]. In practice, we use this mode to compute the divergence of velocity fields parametrized with neural networks. However, isolating the diagonal of the Jacobian to compute the divergence, for general, free-form vector fields, requires $\mathcal{O}(N)$ backward passes [29, 30].

Using a special neural network construction *HollowNet*, Chen and Duvenaud showed, that one can access the full diagonal of the Jacobian, $\mathbf{J} \in \mathbb{R}^{N \times N}$, with a single backward pass. A HollowNet depends on two components:

$$\textbf{Conditioner:} \ h_i = c_i(\mathbf{x}_{\neg i}), \text{where } c_i : \mathbb{R}^{N-1} \to \mathbb{R}^{n_h} \tag{9}$$

$$\textbf{Transformer:} \ b_i = \tau_i(\mathbf{x}_i, \mathbf{h}_i), \text{where } \tau_i : \mathbb{R} \times \mathbb{R}^{n_h} \to \mathbb{R} \tag{10}$$

where $h_i$ is a latent embedding which is independent of the $i$'th dimension of the input, $\mathbf{x}$. Applying the chain rule to $b_i = \tau_i(r_i, c_i(r_{\neg i}))$, the Jacobian $\mathbf{J} = \frac{\mathrm{d}b}{\mathrm{d}\mathbf{x}}$ decomposes as follows:

$$\frac{\mathrm{d}b}{\mathrm{d}\mathbf{x}} = \underbrace{\frac{\partial \tau}{\partial \mathbf{x}}}_{\mathbf{J}^{\text{diag}}} + \underbrace{\frac{\partial \tau}{\partial \mathbf{h}} \frac{\partial \mathbf{h}}{\partial \mathbf{x}}}_{\mathbf{J}^{\text{hollow}}}, \tag{11}$$

which splits the Jacobian into diagonal $\mathbf{J}^{\text{diag}}$ and 'hollow' $\mathbf{J}^{\text{hollow}}$ contributions [29]. The divergence can then be evaluated in $\mathcal{O}(1)$ backward passes, through rewiring the compute graph and computing $\mathbb{1}^T \mathbf{J}^{\text{diag}}$.

# 3 Hollow Message Passing

It is straightforward to construct HollowNets for regular feed-forward neural networks, and convolutional neural networks, through masking operations [29]. However, extending these principles to cases where the data or its domain are subject to other symmetries is not as straightforward.

To address this problem, we develop **Hollow Message Passing** (HoMP) to work with graph data, subject to permutation equivariance. We further specialize this approach to work with data which is subject to common Euclidean symmetries, useful in molecular applications. Here, molecules and particle systems are represented as points in $\mathbb{R}^d$ (typically $d = 3$), one for each of the $n$ atoms or particles, making $N = dn$. HoMP depends on the following developments:

**Permutation Equivariant Conditioner:** We make the conditioner functions $\{c_i\}_{i=1}^n$ permutation equivariant, using a **non-backtracking GNN** (NoBGNN) in combination with a changing underlying graph structure such that it fulfils the same constraints as the original conditioner. Namely, $\frac{\partial h_i}{\partial \mathbf{x}_i} = 0 \ \forall i \in \{1, ..., n\}$.

**Euclidean Symmetries of Node Features:** We lift the HollowNet from working with scalar features, i.e., $\mathbf{x}_i \in \mathbb{R}$ to work with Euclidean features, e.g., $\mathbf{x}_i \in \mathbb{R}^d$. Moreover, NoBGNN allows for the use of message passing architectures that are invariant or equivariant under the group action $\rho(g), g \in \mathcal{G}$ of a group $\mathcal{G}$. Furthermore, for each $i$ there can be additional scalar input data $Z_i \in \mathbb{R}$. In practice (see section 5), $\mathbf{x}_i \in \mathbb{R}^3$ are the coordinates of an atom while $Z_i$ encodes the atom type.

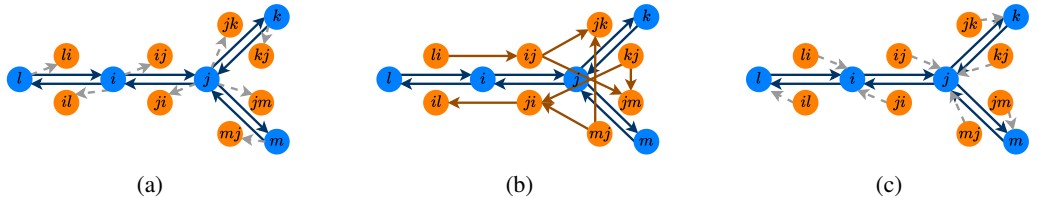

(a)                      (b)                      (c)

Figure 2: **Schematic example of line graph message passing.** Blue: original graph ($G$), orange: line graph ($L(G)$), gray: information flow between $G$ and $L(G)$. (a) Construction of the line graph nodes. Every node in $L(G)$ corresponds to one edge in G. The dashed arrows indicate information flow before the message passing from $G$ to $L(G)$. (b) Construction of the edges of $L(G)$. (c) Information flow after the message passing (dashed arrows) from $L(G)$ to $G$.

**Permutation Equivariant Conditioner**   We design a GNN for the conditioner such that information from any node never returns to itself. A GNN where this holds for one message passing step is called a non-backtracking GNN [31] or a line graph GNN [32] and can be constructed as follows:

Let $G = (N, E)$ be a directed graph with nodes $N := \{1, ..., n\}$ and edges $E \subseteq \{(i, j)|i, j \in N, i \neq j\}$. To each node $i$, we assign a node feature $n_i$. Given $G$, we construct the non-backtracking line graph $L(G) = (N^{lg}, E^{lg})$, where

$$N^{lg} = E, \quad E^{lg} = \{(i, j, k)|(i, j) \in E, (j, k) \in E \text{ and } i \neq k\}. \tag{12}$$

The node features of $L(G)$ are $n_{ij}^{lg}$ are defined to be

$$n_{ij}^{lg} = n_i. \tag{13}$$

The hidden states $h_{ij}^0$ on $L(G)$ are initialized to be $h_{ij}^0 = n_{ij}^{lg}$. Using this setup, a standard message passing scheme [33] is applied on $L(G)$:

$$m_{lij}^t = \phi(h_{li}^t, h_{ij}^t), \quad m_{ij}^t = \sum_{l \in \mathcal{N}^{lg}(i,j)} m_{lij}^t, \quad h_{ij}^{t+1} = \psi(h_{ij}^t, m_{ij}^t), \tag{14}$$

$$b_j = \sum_{i \in \mathcal{N}(j)} R(h_{ij}^{T^{lg}}, n_j) \tag{15}$$

$\phi$, $\psi$ and $R$ are all learnable functions, $\mathcal{N}^{lg}(i, j) = \{k|(k, i, j) \in E^{lg}\}$, $\mathcal{N}(i) = \{k|(k, i) \in E\}$ and $T^{lg}$ is the number of message passing steps on the line graph. This construction is illustrated in fig. 2. The blue nodes and edges are part of $G$ while the orange nodes and edges are part of $L(G)$. The gray dashed arrow in fig. 2(a) illustrate the projection of the node features onto $L(G)$ (eq. (13)). The orange arrows in fig. 2(b) illustrate the message passing on $L(G)$ (eq. (14)). Figure 2(c) visualizes the projection of the node features from $L(G)$ back to $G$ as summarized in eq. (15). Equations (13) and (14) take the role of the conditioner $c(\mathbf{x})$ while eq. (15) takes the role of the transformer $\tau(h, \mathbf{x})$ (see appendix B).

So far, this GNN is, except in special cases that we detail in appendix B, only non-backtracking if just one message passing step is performed. In general, to ensure the non-backtracking nature of the GNN, some connections in $L(G)$ need to be removed after an appropriate number of message passing steps. This can be done by keeping track of which nodes of $L(G)$ received information of which nodes of $G$ after each message passing step. To this end, we define the three dimensional time-dependent backtracking array $B(t) \in \mathbb{R}^{n \times n \times n}$

$$B(t)_{ijk} = \begin{cases} 1 & \text{if } \frac{\partial h_{ij}^t}{\partial \mathbf{x}_k} \neq 0 \\ 0 & \text{else.} \end{cases} \tag{16}$$

with the additional convention that $\frac{\partial h_{ij}^t}{\partial \mathbf{x}_k} = 0$ whenever $(i, j) \notin E$. Notice that whenever $B(t)_{ijj} = 1$ information that started in node $j$ will return to node $j$ during readout. Algorithm 1 shows in detail how $B(t)$ is calculated (steps 8 and 11) and used to remove edges in $L(G)$ (step 10). A more detailed explanation and a proof that this construction does indeed result in a non-backtracking GNN for an arbitrary number of message passing steps is given in appendix B.

**Euclidean Symmetries of Node Features:** In typical molecular applications, nodes are represented in Euclidean space (e.g., $\mathbf{x}_i \in \mathbb{R}^d$ where $d = 3$) and may have additional features $Z_i$ (such as atom type). Our goal is to embed $(\mathbf{x}_i, Z_i) \in \mathbb{R}^d \times \mathbb{R}$ into a set of equivariant ($v_i$) and invariant ($s_i$) node features, $n_i = (v_i, s_i) = \texttt{embed}(\mathbf{x}_i, Z_i) \in \mathbb{R}^{n_h \times d} \times \mathbb{R}^{n_h}$ which serve as inputs for message passing on $L(G)$(see eq. (13)). For this task, we can use any $\mathcal{G}$-equivariant embeddings inherent to GNN architectures such as PaiNN or E3NN (for $d = 3$ and $\mathcal{G} < E(3)$) [26, 27].

The message passing and readout functions (eqs. (14) and (15)) are made $\mathcal{G}$-equivariant by using the message and update functions of any $\mathcal{G}$-equivariant GNN. As we only modify the underlying graph of the GNN, this trivially ensures equivariance.

By construction, the $d$-dimensional node inputs $\mathbf{x}_i$ allow the Jacobian $\mathbf{J}_b$ to be easily split into a block-diagonal and block-hollow part with block size $d \times d$. This observation and its consequences are summarized in algorithm 1 and theorem 1 (see appendix B for a proof and additional explanations). Algorithm 1 can also run on the pairwise euclidean differences of the node features $\mathbf{x}_i$ (pd $= true$ in algorithm 1) . This is common practice in molecular applications to achieve translation invariance. A brief illustration of algorithm 1 can be found in fig. 1.

---

**Algorithm 1** Hollow message passing

---

**Input:** $\{\mathbf{x}_i, Z_i\}_{i=1}^n$ where $\mathbf{x}_i \in \mathbb{R}^d$ and $Z_i \in \mathbb{R}$, $\quad$ pd $\in \{true, false\}$
**Returns:** $\{b_i\}_{i=1}^n$, $b_i \in \mathbb{R}^d$
1: **if** pd **then**
2: $\quad$ $d_{ij} = \mathbf{x}_i - \mathbf{x}_j, \quad e_{ij} = \texttt{embed}(d_{ij}, Z_i)$
3: **else**
4: $\quad$ $n_i = \texttt{embed}(\mathbf{x}_i, Z_i)$
5: **end if**
6: Calculate $G = (N, E)$, e.g., as a $k$NN graph. Self-loops, i.e., $(i, i) \in E$ are prohibited.
7: Calculate the line graph $L(G) = (N^{lg}, E^{lg})$ and its node features $n_{ij}^{lg}$:

$$N^{lg} = E, \ E^{lg} = \{(i, j, k)|(i, j) \in E, (j, k) \in E \text{ and } i \neq k\}, \quad n_{ij}^{lg} = \begin{cases} e_{ij} \text{ if pd} \\ n_i \text{ else} \end{cases}$$

8: Calculate initial node features $h_{ij}^0$ and initial backtracking array $B(0)_{ijk}$:

$$h_{ij}^0 = \begin{cases} \tilde{\psi}( \sum_{k \in \mathcal{N}^{lg}(i,j)} \tilde{\phi}(n_{ki}^{lg})) \text{ if pd} \\ n_{ij}^{lg} \text{ else} \end{cases}, \ B(0)_{ijk} = \begin{cases} (1 \text{ if } k \in \mathcal{N}^{lg}(i,j), \ 0 \text{ else}) \text{ if pd} \\ (1 \text{ if } k = i, \ 0 \text{ else}) \text{ else} \end{cases}$$

9: **for** $t \leftarrow 0$ to $(T^{lg} - 1)$ **do**
10: $\quad$ Remove appropriate edges of $L(G)$: $E^{lg} \leftarrow E^{lg} \setminus \{(i, j, k) \in E^{lg}|B(t)_{ijk} = 1\}$
11: $\quad$ Update backtracking array: $B(t + 1)_{ijk} = \max_{l \in \mathcal{N}^{lg}(i,j)}\{B(t)_{lik}, B(t)_{ijk}\}$
12: $\quad$ Do message passing:
$\quad\quad$ $m_{kij}^t = \phi(h_{ij}^t, h_{ki}^t), \quad m_{ij}^t = \sum_{k \in \mathcal{N}^{lg}(i,j)} m_{kij}^t, \quad h_{ij}^{t+1} = \psi(h_{ij}^t, m_{ij}^t)$
13: **end for**
14: Perform readout and project back to $G$:

$$b_j = \sum_{i \in \mathcal{N}(j)} R(h_{ij}^{T^{lg}}, n_{ij}), \quad \text{where } n_{ij} = \begin{cases} \texttt{embed}(\mathbf{x}_i.\texttt{detach}() - \mathbf{x}_j, Z_i) \quad \text{if pd} \\ n_j \quad \text{else} \end{cases}$$

---

**Theorem 1** (Block-hollowness of HoMP). *Algorithm 1 defines a function $b : \mathbb{R}^{dn} \to \mathbb{R}^{dn}$ whose Jacobian $\mathbf{J}_{b(\mathbf{x})} \in \mathbb{R}^{dn \times dn}$ can be split into a block-hollow and a block-diagonal part, i.e.,*

$$\mathbf{J}_{b(\mathbf{x})} = \mathbf{J}_{c(\mathbf{x})} + \mathbf{J}_{\tau(\mathbf{x})} \tag{17}$$

*where $\mathbf{J}_{c(\mathbf{x})}$ is block-hollow, while $\mathbf{J}_{\tau(\mathbf{x})}$ is block-diagonal with block size $d \times d$, respectively. This structure of the Jacobian enables the exact computation of differential operators that only include diagonal terms of the Jacobian with $d$ instead of $nd$ backward passes through the network.*

In appendix A we show how HoMP can be adapted to attention mechanisms as well.

## 4 Computational Complexity

### 4.1 Graph Connectivity

The number of backward passes through a vector field $b$ parametrized by algorithm 1 that is required to compute its divergence is reduced by a factor of $n$ compared to a standard GNN. However, the forward pass through $b$ is slower than through a standard GNN, as $b$ is operating on the line graph. This overhead will scale with the number of edges, which we denote $\#E$ for $G$ and $\#E^{lg}$ for $L(G)$. In table 1 we show how the number of edges varies with graph types. For a fully connected graph, $\#E^{lg}$ scales as $n^3$ resulting in an impractical overhead in high-dimensional systems. To address this, we consider $k$ nearest-neighbors ($k$NN) graphs instead which scale as $\#E^{lg} = \mathcal{O}(nk^2)$. Choosing $k \leq \mathcal{O}(\sqrt{n})$ will leave the cost of a forward-pass through $b$ comparable to a standard GNN with a fully connected graph. To test whether this restriction in connectivity results in a sufficiently expressive model, we empirically test different values for $k$ in section 5.

Using a $k$NN graph poses a locality assumption that might limit the model's ability to learn the equilibrium distribution of systems with long-range interaction (e.g., systems with coulomb interactions such as molecules). One possible strategy to circumvent this problem is described in appendix C.

Table 1: Number of edges of $L(G)$ and $G$

|            | fully connected    | $k$ neighbors      |
| ---------- | ------------------ | ------------------ |
| $\#E$      | $\mathcal{O}(n^2)$ | $\mathcal{O}(nk)$  |
| $\#E^{lg}$ | $\mathcal{O}(n^3)$ | $\mathcal{O}(nk^2)$ |

### 4.2 Divergence Calculation Speed-up

We summarize our runtime estimations for one integration step during inference $\mathrm{RT}^{step}$, including likelihood calculations, for HollowFlow and a standard GNN based flow model in the following theorem:

**Theorem 2.** *Consider a GNN-based flow model with a fully connected graph $G_{fc}$ and $T$ message passing steps and a HollowFlow model with a $k$ neighbors graph $G_k$ and $T^{lg}$ message passing steps. Let both graphs have $n$ nodes and $d$-dimensional node features.*

*The computational complexity of sampling from the fully connected GNN-based flow model including sample likelihoods is*

$$\mathrm{RT}^{step}(G_{fc}) = \mathcal{O}(Tn^3d), \tag{18}$$

*while the complexity of the HollowFlow model for the same task is*

$$\mathrm{RT}^{step}(L(G_k)) = \mathcal{O}(n(T^{lg}k^2 + dk)). \tag{19}$$

*Moreover, the speed-up of HollowFlow compared to the GNN-based flow model is*

$$\frac{\mathrm{RT}^{step}(G_{fc})}{\mathrm{RT}^{step}(L(G_k))} = \mathcal{O}\left(\frac{Tn^2d}{T^{lg}k^2 + dk}\right). \tag{20}$$

A derivation of these estimates can be found in appendix B.2.

## 5 Experiments

We test HollowFlow on two Lennard-Jones systems with 13 and 55 particles, respectively, (LJ13, LJ55). We also present results on Alanine dipeptide (ALA2) in Appendix C.3.

To this end, we train a CNF (for training details and hyperparameters see appendix D) using the conditional flow matching loss [20] to sample from the equilibrium distribution of these systems. We parametrize the vector field of our CNF using the $O(3)$-equivariant PaiNN architecture [26]. We additionally use minibatch optimal transport [22, 34] and equivariant optimal transport [23]. For each system we train multiple different models:

**Baseline:** $O(3)$-equivariant GNN on a fully connected graph with 3–7 message passing steps.

**HollowFlow**: $O(3)$-equivariant HollowFlow on a $k$NN graph with different values for $k$ with two message passing steps.

In the case of LJ13, we systematically test all values of $k$ from 2 to 12 in steps of 2 to gain insight into how $k$ affects the performance and runtime of the model. Based on our findings and the theoretical scaling laws, we pick $k \in \{7, 27\}$ for LJ55.

Motivated by the symmetry of pairwise interactions between atoms, all $k$NN graphs are symmetrized, i.e., whenever the edge $(i, j)$ is included in the $k$NN graph the edge $(j, i)$ is included as well, even if it is not naturally part of the $k$NN graph.

All training details and hyperparameters are reported in appendix D.

**Criteria to Assess Effectiveness**    We measure the sampling efficiency through the Kish effective samples size (ESS) [35]. As the ESS is highly sensitive to outliers, we remove a right and left percentile of the log importance weights, $\log w_i$, to get a more robust, yet biased, metric, $\text{ESS}_{rem}$ (see appendix D.7) [36]. As HoMP is, in general, a restriction compared to normal message passing, we expect the ESS to be lower. However, as long as the speed-up is sufficiently large, our method can still be an overall improvement.

To gauge the improvement we introduce a compute normalized, relative ESS, i.e., the number of effective samples per GPU second and GPU usage during inference (see eq. (21)). This gives us fairer grounds for comparison of HollowFlow against our baseline, a standard $O(3)$-equivariant CNF. Further, by dividing the relative ESS for all hollow architectures by the baseline relative ESS, we get the effective speed-up (EffSU) and correspondingly the $\text{EffSU}_{rem}$ calculated from the relative $\text{ESS}_{rem}$ in the same way. To ensure a fair comparison, we ran all inference tasks on the same type of GPU for every system, respectively. Details can be found in appendix D.

The results for LJ13 and LJ55 are listed in tables 2 and 3. For both systems we observe that the ESS and the $\text{ESS}_{rem}$ are highest for the baseline and significantly smaller for HollowFlow. However, the relative $\text{ESS}_{rem}$ is significantly higher for all HollowFlow models. This is reflected in the effective speed-up $\text{EffSU}_{rem}$, gauging how many times more samples we can generate per compute compared to our non-hollow baseline. While the maximal $\text{EffSU}_{rem}$ that we observe for LJ13 is about 3.3 for $k = 6$, for LJ55 we observe a maximal $\text{EffSU}_{rem}$ of about 93.7 for $k = 7$. This shows that HollowFlow can reduce the compute required for sampling by about two orders of magnitude on our larger test system.

In fig. 3 we provide a detailed analysis of the runtime of different parts of models trained on LJ13 during sampling. Figure 3(b) demonstrates that the vast majority of the compute in the baseline model is spent on the divergence calculations while only a tiny fraction is spent on the forward pass. fig. 3(a) shows that exactly the opposite is true for HollowFlow, resulting in a decrease of runtime by a factor of up to $\sim 15$, depending on $k$.

$$\text{relative ESS} = \frac{\text{ESS} \times \#\text{samples}}{\text{RT} \times (\text{GPU usage})}, \quad \text{EffSU}(\cdot) = \frac{\text{relative ESS}(\cdot)}{\text{relative ESS}(\text{baseline})} \tag{21}$$

## 6    Related Works

**Efficient Sample Likelihoods in Probability Flow Generative Models**    Evaluating the change in probability associated with probability flow generative models is subject to research but broadly fall into two different categories: 1) approximate estimator by invoking stochastic calculus [37] or trace estimators [30, 38, 39] which are efficient, but prone to high variance 2) NFs with structured Jacobians [16–18, 40, 41], possibly combined with annealing strategies [42–45] which offer efficiency, but limited expressivity or an inability to exploit certain data symmetries such as permutation equivariance.

**Boltzmann Emulators and Other Surrogates**    Several works forego quantitative alignment with a physical potential energy function $u(\mathbf{x})$ in favor of scaling to larger systems through coarse-graining or enabling faster sampling. These qualitative and semi-quantitative Boltzmann surrogates are often

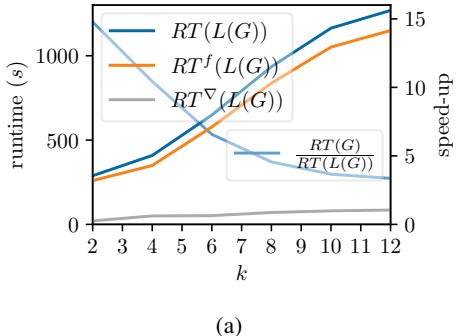
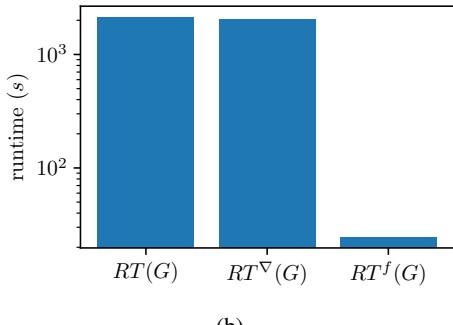

|(a)|(b)|

Figure 3: **Runtimes of HollowFlow and baseline for generating $10^5$ LJ13 equilibrium samples.** All runtimes are multiplied by the GPU usage. (a) Total runtime $RT(L(G))$ and runtime of the forward passes $RT^f(L(G))$ and backward passes $RT^\nabla(L(G))$ of HollowFlow. In light blue: speed-up (ratio of the runtimes of the fully connected baseline $RT(G)$) and HollowFlow . (b) Total runtime and runtime of the forward and backward passes for the non-hollow fully connected baseline. The forward and backward runtimes of the baseline were extrapolated from other runs, see appendix D.6 While the total runtime of the baseline is massively dominated by the backward pass, the opposite is true for HollowFlow.

Table 2: Comparison of HollowFlow trained using different $k$NN graphs and the non-hollow fully connected baseline, all trained on LJ13. The two rightmost columns show the effective speed-up compared to the baseline eq. (21) with and without weight clipping. Bold indicates best performance.

| | ESS (%) | $\text{ESS}_{rem}$ (↑) (%) | $\text{EffSU}_{rem}$ (↑) | EffSU |
|---|---|---|---|---|
| $k = 2$ | $0.054^{0.068}_{0.039}$ | $2.92^{2.95}_{2.89}$ | $1.059^{1.070}_{1.049}$ | $1.16^{1.96}_{0.32}$ |
| $k = 4$ | $0.041^{0.107}_{0.001}$ | $10.34^{10.40}_{10.28}$ | $2.649^{2.667}_{2.631}$ | $0.64^{1.62}_{0.01}$ |
| $k = 6$ | $3.300^{4.301}_{2.410}$ | $20.20^{20.29}_{20.11}$ | $\mathbf{3.260}^{3.278}_{3.243}$ | $31.87^{53.80}_{7.77}$ |
| $k = 8$ | $2.745^{3.236}_{2.250}$ | $20.64^{20.72}_{20.55}$ | $2.310^{2.323}_{2.297}$ | $18.26^{31.12}_{5.65}$ |
| $k = 10$ | $1.057^{1.301}_{0.803}$ | $16.50^{16.58}_{16.41}$ | $1.484^{1.492}_{1.475}$ | $5.51^{9.19}_{1.61}$ |
| $k = 12$ | $4.069^{4.926}_{3.260}$ | $19.72^{19.80}_{19.63}$ | $1.627^{1.636}_{1.619}$ | $20.11^{35.55}_{5.66}$ |
| baseline | $2.132^{2.231}_{0.417}$ | $\mathbf{40.73}^{40.87}_{40.60}$ | $1$ | $1$ |

Table 3: Comparison of HollowFlow trained using different $k$NN graphs and the non-hollow fully connected baseline, all trained on LJ55. The two rightmost columns show the effective speed-up compared to the baseline eq. (21) with and without weight clipping. Bold indicates best performance.

| | ESS (%) | $\text{ESS}_{rem}$ (↑) (%) | $\text{EffSU}_{rem}$ (↑) | EffSU |
|---|---|---|---|---|
| $k = 7$ | $0.006^{0.010}_{0.003}$ | $0.53^{0.56}_{0.51}$ | $\mathbf{93.737}^{99.071}_{88.484}$ | $82.26^{136.56}_{31.04}$ |
| $k = 27$ | $0.007^{0.012}_{0.003}$ | $0.64^{0.68}_{0.61}$ | $9.466^{9.986}_{8.965}$ | $8.46^{14.90}_{3.31}$ |
| $k = 55$ | $0.020^{0.025}_{0.014}$ | $0.74^{0.77}_{0.71}$ | $4.365^{4.583}_{4.144}$ | $9.11^{13.18}_{4.57}$ |
| baseline | $0.048^{0.051}_{0.029}$ | $\mathbf{2.96}^{3.03}_{2.89}$ | $1$ | $1$ |

referred to as Boltzmann Emulators [46]. Some notable examples include, SMA-MD [47], BioEmu [48], AlphaFlow [49], JAMUN [50]. and OpenComplex2 [51]. Other approaches, instead aim to mimic MD simulations with large time-steps including ITO [52–54], TimeWarp [41], ScoreDynamics [55] and related approaches [56–60].

**Non-backtracking Graphs**   Non-backtracking graphs are applied in a variety of classical algorithms, e.g., [61, 62]. NoBGNNs that are non-backtracking for only one message passing step have been applied in a range of other context including for community detection [32], as way to reduce over-squashing [31] and alleviate redundancy in message passing [63]. However, nobody has yet exploited these methods to construct a NoBGNN that is non-backtracking for an arbitrary number of message passing steps to give the Jacobian a structure that allows for cheap divergence calculations.

# 7 Limitations

We demonstrate good theoretical scaling properties of HollowFlow to large systems and support this experimentally on systems of up to 165 dimensions. However, the $k$NN graph used in the current HollowFlow implementation involves an assumption of locality, which may break down in molecular and particle systems with long-range interactions, such as electrostatics. Future work involves exploring strategies to incorporate such long-range interactions, which may involve other strategies to construct the graph used in HollowFlow, Ewald-summation-based methods [64, 65], or linear/fast attention [66, 67]. Also, the fact that some edges in the line graph underlying HollowFlow need to be removed after each message passing step might be problematic at scale. Despite that, there is still a wide range of physical systems where long-range interactions are effectively shielded, such as metals and covalent solids, which might enjoy the properties of HollowFlow models even without these technical advances.

Overall, the performance of our baseline is not in line with state-of-the-art in terms of ESS for the studied systems. However, since our HollowFlow models build directly on the baseline we anticipate any improvement in the baseline to translate into the HollowFlow models as well.

# 8 Conclusions

We introduced HollowFlow, a flow-based generative model leveraging a novel NoBGNN based on Hollow Message Passing (HoMP), enabling sample likelihood evaluations with a constant number of backward passes independent of system size. We demonstrated theoretically that we can achieve sampling speed-ups of up to $\mathcal{O}(n^2)$, if the underlying graph $G$ of HollowFlow is a $k$NN graph. By training a Boltzmann generator on two different systems, we found significant performance gains in terms of compute time for sampling and likelihood evaluation, the largest system enjoying a $10^2 \times$ speed-up in line with our theoretical predictions. Since HoMP can be adopted into any message-passing-based architecture [68], our work can pave the way to dramatically boost the efficiency of expressive CNF-based Boltzmann Generators for high-dimensional systems.

## Acknowledgements

This work was partially supported by the Wallenberg AI, Autonomous Systems and Software Program (WASP) funded by the Knut and Alice Wallenberg Foundation. Results were enabled by resources provided by the National Academic Infrastructure for Supercomputing in Sweden (NAISS) at Alvis (projects: NAISS 2024/22-688 and NAISS 2025/22-841), partially funded by the Swedish Research Council through grant agreement no. 2022-06725. The authors thank Ross Irwin and Selma Moqvist and other team members of the AIMLeNS lab at Chalmers University of Technology for discussions, feedback and comments.

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

# A Adaptation of HollowFlow to Attention

As illustrated in [69], attention can be considered a special case of message passing neural networks. Thus, the adaptation is straightforward, we outline two possible strategies in this section.

First, we replace the general message function $\phi$ (eq. (14)) by a product of the importance coefficients from a self-attention mechanism $a : \mathbb{R}^{n_h} \times \mathbb{R}^{n_h} \to \mathbb{R}$ and a learnable function $\tilde{\phi} : \mathbb{R}^{n_h} \to \mathbb{R}^{n_h}$:

$$m_{kij}^t = a(h_{ij}^t, h_{ki}^t)\tilde{\phi}(h_{ij}^t), \quad m_{ij}^t = \sum_{k \in \mathcal{N}^{lg}(i,j)} m_{kij}^t, \quad h_{ij}^{t+1} = \psi(h_{ij}^t, m_{ij}^t). \tag{22}$$

$\mathcal{N}^{lg}(i,j)$ is defined in eq. (28).

Second, for graph-attention-like mechanism [70], we define the line graph softmax function for a graph $G^q$ with the corresponding notion of nearest neighbors $\mathcal{N}^{lg,q}(i,j)$:

$$\text{softmax}_{ij}^q(y_{ijk}) = \frac{\exp(y_{ijk})}{\sum_{l \in \mathcal{N}^{lg,q}(i,j)} \exp(y_{ijl})}. \tag{23}$$

The updates of HollowFlow then become

$$\alpha_{kij}^{t,q} = \text{softmax}_{ij}^q(a(h_{ij}^t, h_{ki}^t)) \tag{24}$$

$$h_{ij}^{t+1} = \sigma\left(\sum_{k \in \mathcal{N}^{lg,q}(i,j)} \alpha_{kij}^{t,q} \mathbf{W}_{ij} h_{ki}^t\right), \tag{25}$$

where $\mathbf{W}_{ij} \in \mathbb{R}^{n_h \times n_h}$ is a learnable weight matrix, $\sigma : \mathbb{R} \to \mathbb{R}$ is a nonlinearity (applied element wise) and $a : \mathbb{R}^{n_h} \times \mathbb{R}^{n_h} \to \mathbb{R}$ is a non-linear self-attention function, typically a feed-forward neural network. One can also generalize to multi-head attention by independently executing eqs. (24) and (25), possibly even on differently connected graphs $\{G_q\}_{q=1}^H$ potentially circumventing expressiveness problems, and then concatenating the result:

$$\alpha_{kij}^{t,q} = \text{softmax}_{ij}^q(a(\mathbf{W}^q h_{ij}^t, \mathbf{W}^q h_{ki}^t)) \tag{26}$$

$$h_{ij}^{t+1} = \bigg\|_{q=1}^H \sigma\left(\sum_{k \in \mathcal{N}^{lg,q}(i,j)} \alpha_{kij}^{t,q} \mathbf{W}_{ij}^q h_{ki}^t\right). \tag{27}$$

Here, $\mathbf{W}^q$ and $\mathbf{W}_{ij}^q \in \mathbb{R}^{n_h \times n_h \times H}$ are learnable weight matrices and $H$ is the number of attention heads.

These formulations demonstrate that both self-attention and graph-attention layers can be integrated into HollowFlow without altering its block-hollow Jacobian structure, enabling efficient divergence computations within attention-based architectures.

# B Derivations and Proofs

## B.1 Hollowness of HollowFlow

We first reintroduce some notation and do then restate algorithm 1 and theorem 1 and proof it.

**Notational remarks**  By $\mathcal{N}^{lg}(i,j)$ and $\mathcal{N}(i)$ we mean the following notions of nearest neighbors:

$$\mathcal{N}^{lg}(i,j) = \{k|(k,i,j) \in E^{lg}\} \tag{28}$$
$$\mathcal{N}(i) = \{k|(k,i) \in E\}. \tag{29}$$

Equation (28) is illustrated in fig. 4. The functions $\phi, \tilde{\phi}, \psi, \tilde{\psi}$ and $R$ used in algorithm 1 are all learnable functions. $B(t) \in \mathbb{R}^{n \times n \times n}$ is the three-dimensional time-dependent back tracking array (see eq. (16) and algorithm 1), the integer $t$ indicates the current message passing step. The binary variable pd $\in \{true, false\}$ indicated whether the algorithms runs on the (translation invariant) pairwise differences or not. The detach() method means that derivatives w.r.t. the detached variable are ignored (i.e., equal to zero).

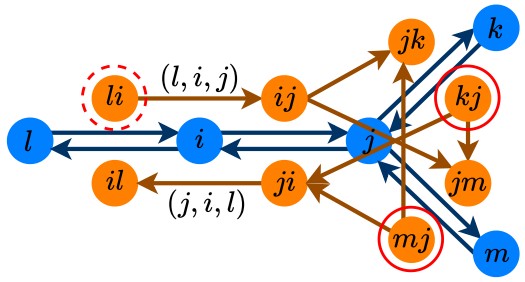

Figure 4: **Schematic example of the graph** $G$ **(in blue) and its line graph** $L(G)$ **(in orange):** The line graph nodes encircled with a continuos red line correspond to $\mathcal{N}^{lg}(j, i)$ while the node encircled with a dashed red line corresponds to $\mathcal{N}^{lg}(i, j)$. Two edges of $L(G)$ are labelled explicitly to illustrate our labelling convention.

---

**Algorithm 1** Hollow message passing

---

**Input:** $\{\mathbf{x}_i, Z_i\}_{i=1}^n$ where $\mathbf{x}_i \in \mathbb{R}^d$ and $Z_i \in \mathbb{R}$,   pd $\in \{true, false\}$
**Returns:** $\{b_i\}_{i=1}^n$, $b_i \in \mathbb{R}^d$
1: **if** pd **then**
2:     $d_{ij} = \mathbf{x}_i - \mathbf{x}_j$,   $e_{ij} = \texttt{embed}(d_{ij}, Z_i)$
3: **else**
4:     $n_i = \texttt{embed}(\mathbf{x}_i, Z_i)$
5: **end if**
6: Calculate $G = (N, E)$, e.g., as a $k$NN graph. Self-loops, i.e., $(i, i) \in E$ are prohibited.
7: Calculate the line graph $L(G) = (N^{lg}, E^{lg})$ and its node features $n_{ij}^{lg}$:

$$N^{lg} = E, \ E^{lg} = \{(i, j, k) | (i, j) \in E, (j, k) \in E \text{ and } i \neq k\}, \quad n_{ij}^{lg} = \begin{cases} e_{ij} \text{ if pd} \\ n_i \text{ else} \end{cases}$$

8: Calculate initial node features $h_{ij}^0$ and initial backtracking array $B(0)_{ijk}$:

$$h_{ij}^0 = \begin{cases} \tilde{\psi}(\sum_{k \in \mathcal{N}^{lg}(i,j)} \tilde{\phi}(n_{ki}^{lg})) \text{ if pd} \\ n_{ij}^{lg} \text{ else} \end{cases}, \quad B(0)_{ijk} = \begin{cases} (1 \text{ if } k \in \mathcal{N}^{lg}(i, j), \ 0 \text{ else}) \text{ if pd} \\ (1 \text{ if } k = i, \ 0 \text{ else}) \text{ else} \end{cases}$$

9: **for** $t \leftarrow 0$ to $(T^{lg} - 1)$ **do**
10:     Remove appropriate edges of $L(G)$: $E^{lg} \leftarrow E^{lg} \setminus \{(i, j, k) \in E^{lg} | B(t)_{ijk} = 1\}$
11:     Update backtracking array: $B(t + 1)_{ijk} = \max_{l \in \mathcal{N}^{lg}(i,j)}\{B(t)_{lik}, B(t)_{ijk}\}$
12:     Do message passing:
    $m_{kij}^t = \phi(h_{ij}^t, h_{ki}^t)$,   $m_{ij}^t = \sum_{k \in \mathcal{N}^{lg}(i,j)} m_{kij}^t$,   $h_{ij}^{t+1} = \psi(h_{ij}^t, m_{ij}^t)$
13: **end for**
14: Perform readout and project back to $G$:

$$b_j = \sum_{i \in \mathcal{N}(j)} R(h_{ij}^{T^{lg}}, n_{ij}), \quad \text{where } n_{ij} = \begin{cases} \texttt{embed}(\mathbf{x}_i.\texttt{detach}() - \mathbf{x}_j, Z_i) & \text{if pd} \\ n_j & \text{else} \end{cases}$$

---

**Theorem 1** (Block-hollowness of HoMP). *Algorithm 1 defines a function* $b : \mathbb{R}^{dn} \rightarrow \mathbb{R}^{dn}$ *whose Jacobian* $\mathbf{J}_{b(\mathbf{x})} \in \mathbb{R}^{dn \times dn}$ *can be split into a block-hollow and a block-diagonal part, i.e.,*

$$\mathbf{J}_{b(\mathbf{x})} = \mathbf{J}_{c(\mathbf{x})} + \mathbf{J}_{\tau(\mathbf{x})} \tag{30}$$

*where* $\mathbf{J}_{c(\mathbf{x})}$ *is block-hollow, while* $\mathbf{J}_{\tau(\mathbf{x})}$ *is block-diagonal with block size* $d \times d$*, respectively. This structure of the Jacobian enables the exact computation of differential operators that only include diagonal terms of the Jacobian with* $d$ *instead of* $nd$ *backward passes through the network.*

*Remark:* Figure 4 might be a helpful illustration to keep track of the indices when reading the proof.

*Proof.* By the chain rule step 14 gives

$$\frac{db_j}{d\mathbf{x}_k} = \sum_{i \in \mathcal{N}(j)} \left( \frac{\partial R}{\partial h_{ij}^{T^{lg}}} \frac{\partial h_{ij}^{T^{lg}}}{\partial \mathbf{x}_k} + \frac{\partial R}{\partial n_{ij}} \frac{\partial n_{ij}}{\partial \mathbf{x}_k} \right). \tag{31}$$

If we show that $\frac{\partial}{\partial \mathbf{x}_j} h_{ij}^t = 0 \ \forall t \geq 0$ and $\left( \frac{\partial n_{ij}}{\partial \mathbf{x}_k} = 0 \text{ if } j \neq k \right)$ for all $(i, j) \in E$ and for all $k \in \{1, ..., n\}$ the Jacobian decomposes as stated. We split the proof into two cases:

**Case 1: pd $= false$:**

That $\frac{\partial n_{ij}}{\partial \mathbf{x}_k} = 0$ if $j \neq k$ is trivial for all $k \in \{1, ..., n\}$ as $n_{ij} = \texttt{embed}(\mathbf{x}_j, Z_j)$.

We show $\frac{\partial}{\partial \mathbf{x}_j} h_{ij}^t = 0$ for all $t \geq 0$ and for all $(i, j) \in E$ by induction on $t$:

**Base case:**

1. $h_{ij}^0 = \texttt{embed}(\mathbf{x}_i, Z_i)$ is by definition independent of $\mathbf{x}_j$ for all $(i, j) \in E$.

2. Furthermore, $B(0)_{ijk}$ is by definition one exactly when $h_{ij}^0$ depends on $\mathbf{x}_k$ and zero otherwise (see step 8).

**Induction step:** Assume

1. $h_{ij}^t$ is independent of $\mathbf{x}_j$ for all $(i, j) \in E$ and

2. $B(t)_{ijk}$ is zero exactly when $h_{ij}^t$ is independent of $\mathbf{x}_k$.

We want to show that these conditions also hold for $t + 1$.

To see that $h_{ij}^{t+1}$ is independent of $\mathbf{x}_j$ for all $(i, j) \in E$, by step 12 of algorithm 1 and the first induction hypothesis, we only need to show that $h_{ki}^t$ does not depend on $\mathbf{x}_j$ for all $k \in \mathcal{N}^{lg}(i, j)$. Assume $h_{ki}^t$ does depend on $\mathbf{x}_j$. By the second induction hypothesis, this implies $B(t)_{kij} = 1$ and by step 10 of algorithm 1 thus $(k, i, j) \notin E^{lg}$. This means $k \notin \mathcal{N}^{lg}(i, j)$, a contradiction.

Next, we prove $(B(t + 1)_{ijk} = 0$ if and only if $h_{ij}^{t+1}$ is independent of $\mathbf{x}_k)$ by proving both implications:

$B(t + 1)_{ijk} = 0 \implies h_{ij}^{t+1}$ is independent of $\mathbf{x}_k$:
Assume $B_{ijk}(t + 1) = 0$. By step 11 of algorithm 1 this implies $B(t)_{lik} = 0 \ \forall l \in \mathcal{N}^{lg}(i, j)$ and $B(t)_{ijk} = 0$. By the second induction hypothesis it follows that $h_{li}^t$ and $h_{ij}^t$ are independent of $\mathbf{x}_k$ for all $l \in \mathcal{N}^{lg}(i, j)$ which by step 12 of algorithm 1 implies that $h_{ij}^{t+1}$ is independent of $\mathbf{x}_k$.

$h_{ij}^{t+1}$ is independent of $\mathbf{x}_k \implies B(t + 1)_{ijk} = 0$:
Assume $h_{ij}^{t+1}$ is independent of $\mathbf{x}_k$. By step 12 of algorithm 1 this implies that $h_{ij}^t$ and $h_{li}^t$ are independent of $\mathbf{x}_k$ for all $l \in \mathcal{N}^{lg}(i, j)$. By the second induction hypothesis and step 11 of algorithm 1 we get $B(t + 1)_{ijk} = 0$ as desired.

**Case 2: pd $= true$:**

That $\frac{\partial n_{ij}}{\partial \mathbf{x}_k} = 0$ if $j \neq k$ is trivial for all $k \in \{1, ..., n\}$ as $n_{ij} = \texttt{embed}(\mathbf{x}_i.\texttt{detach}() - \mathbf{x}_j, Z_i))$ and the $\texttt{detach}()$ method means that we ignore derivatives w.r.t. the detached variable.

We again show $\frac{\partial}{\partial \mathbf{x}_j} h_{ij}^t = 0$ for all $t \geq 0$ and for all $(i, j) \in E$ by induction on $t$. The only difference compared to case 1 is the base case, as $h_{ij}^0$ is initialized differently.

**Base case:**

1. $h_{ij}^0$ is a function of $n_{ki}^{lg}$ where $k \in \mathcal{N}^{lg}(i,j)$. As $k \neq j$ and $i \neq j$, $h_{ij}^0$ is independent of $\mathbf{x}_j$.

2. We need to show $B(0)_{ijk}$ is zero exactly when $h_{ij}^0$ is independent of $\mathbf{x}_k$. Assume $B(0)_{ijk} = 0$. Then $k \notin \mathcal{N}^{lg}(i,j)$ by step 8. Thus, by definition of $h_{ij}^0$ (step 8), $h_{ij}^0$ is independent of $\mathbf{x}_k$. Conversely, let $h_{ij}^0$ be independent of $\mathbf{x}_k$. Then $k \notin \mathcal{N}^{lg}(i,j)$ meaning $B(0)_{ijk} = 0$ by step 8.

**Induction step:**   The induction step is the same as in case 1.

**Jacobian diagonal with $d$ backward passes:**   We can now exploit the decomposition eq. (31) of the Jacobian to get all its diagonal terms with only $d$ vector-Jacobian products (backward passes). This can be achieved by applying the `detach()` operation to $h_{ij}^{T^{lg}}$ in step 14 of algorithm 1 such that the block-diagonal part of the decomposition eq. (31) remains unchanged while the hollow part vanishes. If we construct $d$ column vectors $\{v_i \in \mathbb{R}^{dn}\}_{i=1}^d$ as

$$(v_i)_j = \begin{cases} 1 & \text{if } j = i + d(k-1), \ k \in \{1, ..., n\} \\ 0 & \text{else,} \end{cases} \tag{32}$$

The vector-Jacobian products $\{\mathbf{J}_b v_i\}_{i=1}^d$ do then give us access to all diagonal terms of $\mathbf{J}_b$, e.g., $\mathrm{tr}(\mathbf{J}_b) = \sum_{i=1}^d v_i^T \mathbf{J}_b v_i$ (see fig. 1 for an illustration in the case $n = 5, d = 3$).  $\square$

**Further remarks**   The conditioner $c(\mathbf{x})$ and the transformer $\tau(h, \mathbf{x})$ shown in fig. 1 can explicitly be written component-wise as follows:

$$c_j(\mathbf{x}) = \{h_{ij}^{T^{lg}}\}_{i \in \mathcal{N}(j)} := h_j \tag{33}$$

$$\tau_j(h_j, \mathbf{x}_j) = \sum_{i \in \mathcal{N}(j)} R(h_{ij}^{T^{lg}}, n_{ij}), \tag{34}$$

where $h_{ij}^{T^{lg}}$ and $n_{ij}$ are defined as in algorithm 1.

Whenever a directed graph $G$ acyclic, its directed line graph $L(G)$ will also be acyclic which implies that even without edge removal in $L(G)$ (step 10 in algorithm 1) message passing on the line graph ensures that information starting from node $i$ of $G$ can never return to itself after an arbitrary number of message passing steps. If the shortest cycle in $G$ has length $g$ (i.e., $G$ has *girth $g$*), one can do at most $g - 2$ message passing steps without returning information if no edges of $L(G)$ are removed. The shift $-2$ comes from the fact that initialization of the node features (step 7 in algorithm 1) and the readout (step 14 in algorithm 1) effectively bridge one edge of $G$ each.

## B.2   Computational Complexities

**Theorem 2.** *Consider a GNN-based flow model with a fully connected graph $G_{fc}$ and $T$ message passing steps and a HollowFlow model with a $k$ neighbors graph $G_k$ and $T^{lg}$ message passing steps. Let both graphs have $n$ nodes and $d$-dimensional node features.*

*The computational complexity of sampling from the fully connected GNN-based flow model including sample likelihoods is*

$$\mathrm{RT}^{step}(G_{fc}) = \mathcal{O}(Tn^3 d), \tag{18}$$

*while the complexity of the HollowFlow model for the same task is*

$$\mathrm{RT}^{step}(L(G_k)) = \mathcal{O}(n(T^{lg}k^2 + dk)). \tag{19}$$

*Moreover, the speed-up of HollowFlow compared to the GNN-based flow model is*

$$\frac{\mathrm{RT}^{step}(G_{fc})}{\mathrm{RT}^{step}(L(G_k))} = \mathcal{O}\left(\frac{Tn^2 d}{T^{lg}k^2 + dk}\right). \tag{20}$$

*Proof.* The sampling procedure involves a forward pass and one divergence calculation for each integration step. We start by estimating the runtimes of the forward pass $\text{RT}^f$. The key assumption is that the computational complexity of the forward pass is proportional to the number of edges of the graph (see table 1) and the number of message passing steps.:

$$\text{RT}^f(G_{fc}) = \mathcal{O}(T\#E_{fc}) = \mathcal{O}(Tn^2) \tag{35}$$

$$\text{RT}^f(L(G_k)) = \mathcal{O}(T^{lg}\#E_k^{lg}) = \mathcal{O}(T^{lg}nk^2). \tag{36}$$

The computational complexity of the divergence calculation $RT^\nabla$ will be proportional to the number of backward passes necessary to compute the divergence times the complexity for each backward pass. For the fully connected GNN we need $dn$ backward passes at the cost of $\mathcal{O}(T\#E_{fc})$:

$$\text{RT}^\nabla(G_{fc}) = \mathcal{O}(T\#E_{fc}dn) = \mathcal{O}(Tn^3d). \tag{37}$$

For HollowFlow we only need $d$ backward passes by exploiting the structure of the Jacobian (See theorem 1). Each of these backward passes has a complexity $\mathcal{O}(\#E_k)$, as we only need to backpropagate through the transformer, not the conditioner:

$$\text{RT}^\nabla(L(G_k)) = \mathcal{O}(\#E_kd) = \mathcal{O}(dnk). \tag{38}$$

The runtime of one integration step, $\text{RT}^{step}$, that is proportional to the total runtime, will scale as $\text{RT}^f + \text{RT}^\nabla$. Thus, we get

$$\text{RT}^{step}(G_{fc}) = \text{RT}^f(G_{fc}) + \text{RT}^\nabla(G_{fc}) = \mathcal{O}(Tn^2) + \mathcal{O}(Tn^3d) = \mathcal{O}(Tn^3d), \tag{39}$$

$$\text{RT}^{step}(L(G_k)) = \text{RT}^f(L(G_k)) + \text{RT}^\nabla(L(G_k)) = \mathcal{O}(T^{lg}nk^2) + \mathcal{O}(dnk) = \mathcal{O}(n(T^{lg}k^2 + dk)), \tag{40}$$

where the right term in the sum in eq. (39) dominates as $\mathcal{O}(n^3) > \mathcal{O}(n^2)$ and $d$ is a constant independent of $n$ and $T$. The speed-up during sampling with likelihood evaluation when using HollowFlow compared to a standard, fully connected GNN is thus

$$\frac{\text{RT}^{step}(G_{fc})}{\text{RT}^{step}(L(G_k))} = \mathcal{O}\left(\frac{Tn^2d}{T^{lg}k^2 + dk}\right) = \mathcal{O}\left(\frac{Tnd}{T^{lg}}\right), \tag{41}$$

where the last equality assumes $k = \mathcal{O}(\sqrt{n})$ (see section 4.1). In conclusion, we expect a speed-up of $\mathcal{O}(n^2)$ for constant $k$ and a speed-up of $\mathcal{O}(n)$ for $k = \mathcal{O}(\sqrt{n})$. $\qquad\square$

### B.3 Memory scaling of $B(t)$

From the definition of $B(t)$ (eq. (16)) one might expect a cubic memory scaling in $n$. However, it is sufficient to only save information about from which nodes of $G$ all the nodes of $L(G)$ have received information from. If $G$ has $n$ nodes and $L(G)$ has $n_{lg}$ nodes, the size of the array will effectively be $nn_{lg}$. For a $k$NN graph, $n_{lg} = \mathcal{O}(nk)$ so the memory requirements of $B(t)$ in our implementation scale as $n^2k$.

## C  Additional Results

### C.1  Non-equivariant HollowFlow

Adaptation of HollowFlow to non-equivariant GNN architectures is straightforward as HollowFlow only affects the construction of the underlying graph while preserving the message and update functions of the GNN including their equivariance properties. Relaxing permutation symmetry can be done by using unique embeddings for all nodes in the graph (as done for Alanine Dipeptide, see appendix C.3).

### C.2  Multi-head HollowFlow

Using a $k$NN graph poses a locality assumption that might limit the models ability when learning the distribution of systems with long-range interactions (e.g., systems with coulomb interactions such as molecules). To circumvent the locality assumption while still keeping the computational cost manageable, we additional consider a multi-head strategy with $H$ heads. The central idea is to run HoMP on $H$ different graphs in parallel and sum up the result. These graphs can be constructed in different ways, we explore the following two strategies:

| | ESS (%) | $\text{ESS}_{rem}$ (↑) (%) | $\text{EffSU}_{rem}$ (↑) | EffSU |
|---|---|---|---|---|
| $k=2$ | $0.054^{0.068}_{0.039}$ | $2.92^{2.95}_{2.89}$ | $1.059^{1.070}_{1.049}$ | $1.16^{1.96}_{0.32}$ |
| $k=4$ | $0.041^{0.107}_{0.001}$ | $10.34^{10.40}_{10.28}$ | $2.649^{2.667}_{2.631}$ | $0.64^{1.62}_{0.01}$ |
| $k=6$ | $3.300^{4.301}_{2.410}$ | $20.20^{20.29}_{20.11}$ | $\mathbf{3.260}^{3.278}_{3.243}$ | $31.87^{53.80}_{7.77}$ |
| $k=8$ | $2.745^{3.236}_{2.250}$ | $20.64^{20.72}_{20.55}$ | $2.310^{2.323}_{2.297}$ | $18.26^{31.12}_{5.65}$ |
| $k=10$ | $1.057^{1.301}_{0.803}$ | $16.50^{16.58}_{16.41}$ | $1.484^{1.492}_{1.475}$ | $5.51^{9.19}_{1.61}$ |
| $k=12$ | $4.069^{4.926}_{3.260}$ | $19.72^{19.80}_{19.63}$ | $1.627^{1.636}_{1.619}$ | $20.11^{35.55}_{5.66}$ |
| $H=2$ | $2.199^{2.604}_{1.783}$ | $17.55^{17.64}_{17.47}$ | $1.613^{1.622}_{1.604}$ | $12.08^{20.58}_{3.51}$ |
| $H=3$ | $0.741^{0.932}_{0.547}$ | $13.78^{13.85}_{13.71}$ | $1.339^{1.347}_{1.331}$ | $4.17^{6.98}_{1.11}$ |
| $H=4$ | $0.333^{0.742}_{0.054}$ | $11.32^{11.38}_{11.26}$ | $1.064^{1.071}_{1.057}$ | $1.85^{4.17}_{0.17}$ |
| $H=2, I=1$ | $3.730^{5.204}_{2.466}$ | $23.35^{23.44}_{23.25}$ | $0.864^{0.868}_{0.859}$ | $8.53^{14.71}_{1.98}$ |
| $H=3, I=1$ | $2.484^{3.442}_{1.626}$ | $19.06^{19.14}_{18.98}$ | $1.206^{1.213}_{1.200}$ | $9.23^{15.60}_{2.34}$ |
| $H=4, I=1$ | $0.322^{0.408}_{0.189}$ | $11.52^{11.58}_{11.46}$ | $0.845^{0.851}_{0.840}$ | $1.42^{2.33}_{0.35}$ |
| $H=4, I=2$ | $2.239^{2.541}_{1.934}$ | $15.32^{15.40}_{15.25}$ | $0.849^{0.854}_{0.844}$ | $7.40^{12.90}_{2.24}$ |
| baseline $k=6$ | $0.002^{0.003}_{0.001}$ | $\mathbf{43.52}^{43.63}_{43.41}$ | $1.010^{1.014}_{1.005}$ | $0.00^{0.00}_{0.00}$ |
| baseline | $2.132^{2.231}_{0.417}$ | $40.73^{40.87}_{40.60}$ | $1$ | $1$ |

1. **Non-overlapping scale separation**: Consider a fully connected graph $G$ embedded in euclidean space with $\#E$ edges and sort the edges by length. Divide this list into $H$ equally sized chunks of length $\#E/H$ with no overlap. If there is a remainder, distribute the remaining edges to the heads as equally as possible.

2. **Overlapping scale separation with overlap number $I$**: Consider a fully connected graph $G$ embedded in euclidean space with $\#E$ edges and sort the edges by length. Divide this list into $H$ equally sized chunks of size $\#E/(H-I)$ such that the centers of these chunks are spaced evenly. If there is a remainder, distribute the remaining edges to the heads as equally as possible.

Both strategies ensure that there is no locality assumption. The first strategy assumes scale separability while the second one allows for a (limited) scale overlap. In terms of scaling, observe that $\#E = \mathcal{O}(n^2)$ for a fully connected graph. Thus, each head has $\mathcal{O}(n^2/(H-I))$ edges giving us on average $\mathcal{O}(n/(H-I))$ edges per node. Finally, the line graph of each head has thus $\mathcal{O}(n^3/(H-I)^2)$ edges, giving us $\mathcal{O}(Hn^3/(H-I)^2)$ line graph edges in total. Thus, if we, e.g., choose the number of heads $H$ and the overlap number $I$ proportional to $n$, the scaling is $\mathcal{O}(n^2)$.

We test both of the aforementioned multi-head strategies on LJ13, LJ55 and Alanine Dipeptide. The results together with the single-head results from the main paper can be found in tables 4 to 6.

## C.3   Alanine Dipeptide

We additionally trained HollowFlow and a corresponding baseline on Alanine Dipeptide using a number of different graph connecting strategies (multi headed and single headed). For all models, we break permutation equivariance. Interestingly, the overlapping multi-head strategy with $H=3$ and $I=1$ seems to perform best in terms of $\text{ESS}_{rem}$ out of all HollowFlow models, suggesting that multi-head approaches might be a promising direction to alleviate limitations of HollowFlow. However, we do not observe an effective speed-up of HollowFlow compared to the non-hollow baseline. We believe that further hyperparameter tuning and engineering efforts might close this performance gap.

Table 5: LJ55

| | ESS (%) | ESS$_{rem}$ ($\uparrow$) (%) | EffSU$_{rem}$ ($\uparrow$) | EffSU |
|---|---|---|---|---|
| $k = 7$ | $0.006^{0.010}_{0.003}$ | $0.53^{0.56}_{0.51}$ | $\mathbf{93.737}^{99.071}_{88.484}$ | $82.26^{136.56}_{31.04}$ |
| $k = 27$ | $0.007^{0.012}_{0.003}$ | $0.64^{0.68}_{0.61}$ | $9.466^{9.986}_{8.965}$ | $8.46^{14.90}_{3.31}$ |
| $k = 55$ | $0.020^{0.025}_{0.014}$ | $\mathbf{0.74}^{0.77}_{0.71}$ | $4.365^{4.583}_{4.144}$ | $9.11^{13.18}_{4.57}$ |
| $H = 2$ | $0.008^{0.011}_{0.003}$ | $0.53^{0.56}_{0.50}$ | $5.259^{5.580}_{4.946}$ | $6.44^{9.07}_{2.23}$ |
| $H = 4$ | $0.006^{0.009}_{0.003}$ | $0.71^{0.74}_{0.67}$ | $7.198^{7.576}_{6.819}$ | $4.86^{8.05}_{1.85}$ |
| $H = 6$ | $0.009^{0.013}_{0.005}$ | $0.48^{0.51}_{0.45}$ | $5.901^{6.244}_{5.551}$ | $9.39^{13.89}_{3.88}$ |
| $H = 8$ | $0.015^{0.020}_{0.010}$ | $0.53^{0.55}_{0.50}$ | $7.144^{7.569}_{6.712}$ | $15.41^{23.41}_{6.96}$ |
| $H = 10$ | $0.006^{0.010}_{0.003}$ | $0.53^{0.56}_{0.50}$ | $7.604^{8.049}_{7.153}$ | $7.02^{11.52}_{2.90}$ |
| baseline $k = 27$ | $0.007^{0.012}_{0.003}$ | $0.64^{0.67}_{0.61}$ | $0.324^{0.342}_{0.307}$ | $0.28^{0.48}_{0.11}$ |
| baseline | $0.048^{0.051}_{0.029}$ | $2.96^{3.03}_{2.89}$ | $1$ | $1$ |

Table 6: ALA2

| | ESS (%) | ESS$_{rem}$ ($\uparrow$) (%) | EffSU$_{rem}$ ($\uparrow$) | EffSU |
|---|---|---|---|---|
| $H = 2$ | $0.008^{0.011}_{0.005}$ | $0.68^{0.71}_{0.65}$ | $0.566^{0.589}_{0.544}$ | $3.63^{5.56}_{0.27}$ |
| $H = 3$ | $0.012^{0.017}_{0.007}$ | $0.23^{0.25}_{0.22}$ | $0.198^{0.213}_{0.183}$ | $4.97^{10.09}_{0.42}$ |
| $H = 4$ | $0.006^{0.008}_{0.003}$ | $0.11^{0.13}_{0.09}$ | $0.114^{0.128}_{0.102}$ | $2.71^{5.56}_{0.22}$ |
| $H = 5$ | $0.003^{0.005}_{0.002}$ | $0.04^{0.05}_{0.03}$ | $0.042^{0.050}_{0.034}$ | $1.43^{3.24}_{0.11}$ |
| $H = 3, I = 1$ | $0.017^{0.020}_{0.013}$ | $0.73^{0.76}_{0.69}$ | $0.399^{0.416}_{0.382}$ | $3.64^{7.00}_{0.42}$ |
| $k = 11$ | $0.010^{0.012}_{0.006}$ | $0.52^{0.54}_{0.50}$ | $0.588^{0.618}_{0.550}$ | $4.52^{8.59}_{0.29}$ |
| baseline | $0.135^{0.224}_{0.019}$ | $\mathbf{16.14}^{16.29}_{15.97}$ | $\mathbf{1}$ | $1$ |

# D  Experimental Details

## D.1  Code and Libraries

All models and training is implemented using *PyTorch* [71] with additional use of the following libraries: *bgflow* [1, 24], *torchdyn* [72], *TorchCFM* [20, 22] and *SchNetPack* [73, 74]. The conditional flow matching and equivariant optimal transport implementation is based on the implementation used in [23]. The implementation of the PaiNN [26] architecture is based on [73, 74]. The code and the models are available here: `https://github.com/olsson-group/hollowflow`.

## D.2  Benchmark Systems

The potential energy of the Lennard-Jones systems LJ13 and LJ55 is given by

$$U^{LJ}(\mathbf{x}) = \frac{\epsilon}{2\tau} \left[ \sum_{i,j} \left( \left( \frac{r_m}{d_{ij}} \right)^{12} - 2 \left( \frac{r_m}{d_{ij}} \right)^{6} \right) \right], \tag{42}$$

where the parameters are as in [23]: $r_m = 1$, $\epsilon = 1$ and $\tau = 1$. $d_{ij}$ is the pairwise euclidean distance between particle $i$ and $j$. Detailed information about the molecular system (Alanine Dipeptide) can be found in [23].

## D.3  Training Data

We used the same training data as in [23] available at `https://osf.io/srqg7/?view_only=28deeba0845546fb96d1b2f355db0da5`. The training data has been generated using MCMC, details can be found in the appendix of [23]. For all systems we used $10^5$ randomly selected samples for training as in [23]. Validation was done using $10^4$ of the remaining samples.

### D.4 Hyperparameters

**Choice of the Hyperparameter** $k$    Heuristically, we expect maximal expressiveness for mid-range values of $k$: By construction, some of the connections in the line graph need to be removed after a certain number of message passing (MP) steps (See appendix B for more details). The larger $k$, the more connections need to be removed after only one MP step up to the point where (almost) no connections are left after only one MP step. We thus have two counteracting effects when $k$ increases: The first MP step gains more connections and thus expressiveness while the second MP step and all later MP steps loose connections and expressiveness. This effect is illustrated in fig. 5. Eventually, the latter seems to take over limiting the expressiveness of the model for large values of $k$. This non-linear qualitative scaling of performance with $k$ is experimentally supported by the results in table 2.

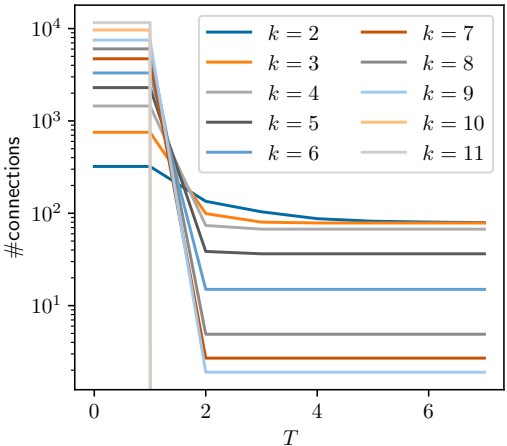

Figure 5: Average number of connections left in the line graph of a $k$NN graph $G$ as a function of the number of message passing steps. $G$ has 55 nodes whose coordinates are sampled from a standard Gaussian. The average is taken over 10 independently sampled graphs. The larger $k$, the faster the line graph disconnects.

**Discontinuities Caused by** $k$**NN Graph**    The vector field $b_\theta(x)$ (eq. (3)) is constructed using a $k$NN graph. This might cause discontinuities on the decision boundaries of the $k$NN construction. As the divergence of the vector field is only defined if it is differentiable, these discontinuities can cause theoretical issues, including but not limited to the continuity equation not being well-defined everywhere eq. (4). We have not observed any numerical instabilities or other issues in practice. To further validate empirically that the discontinuities do not cause problems, we tracked the divergence of a trained, fully connected as well as a trained, 6 nearest-neighbors HollowFlow LJ13 model during sampling using a forward Euler scheme with 20 integration steps. We monitor the maximum and average absolute difference of the divergence of consecutive steps, computed over 1000 initial conditions, to assess whether there are significant discontinuities in the divergence of the nearest-neighbors model compared to the fully connected one. Results can be found in table 7. While both values are slightly higher for the $k$NN model, there are no significant differences further supporting our claim that the discontinuities do not cause issues in practice.

Table 7: Maximal and average absolute difference of the divergence of consecutive steps, computed over 1000 initial conditions, for two different LJ13 HollowFlow models.

| model | maximal difference | average difference |
|---|---|---|
| fully connected | 72.5 | 5.3 |
| $k = 6$ | 74.2 | 5.8 |

**Hyperparameters Used in Experiments:**  Depending on the system, we used a different number of message passing steps and a different value for the hidden dimension $n_h$, following [23]. For LJ13, we chose $n_h = 32$ for all runs. All HollowFlow experiments with LJ13 and $k$NN graphs used two message passing steps while the baselines and the multi-head experiments used three. For LJ55, we chose $n_h = 64$ for all runs. All HollowFlow experiments with LJ55 and $k$NN graphs used two message passing steps while the baselines and the multi-head experiments used seven. For Alanine Dipeptide, we chose $n_h = 64$ and five message passing steps for all runs. All neural networks in the PaiNN architecture use the *SiLU* activation function [75].

The training details for LJ13, LJ55 and Alanine Dipeptide are reported in tables 8 to 10. For LJ13, we selected the last model for all $k$NN experiments and the fully connected baseline experiment, while we selected the model with the lowest validation loss for all remaining experiments for inference. For LJ55 and Alanine Dipeptide we always selected the model with the lowest validation loss for inference. All training was done using the *Adam* optimizer [76].

Generally, we observed that HollowFlow can be trained with a larger learning rate compared to the baseline without running into instability issues. Nevertheless, the effect on model performance seemed limited.

Table 8: Training details LJ13

| | batch size | learning rate | epochs | training time (h) |
|---|---|---|---|---|
| $k = 2$ | 1024 | $5 \times 10^{-4}$ | 1000 | 5.8 |
| $k = 4$ | 1024 | $5 \times 10^{-4}$ | 1000 | 6.3 |
| $k = 6$ | 1024 | $5 \times 10^{-4}$ | 1000 | 6.7 |
| $k = 8$ | 1024 | $5 \times 10^{-4}$ | 1000 | 7.3 |
| $k = 10$ | 1024 | $5 \times 10^{-4}$ | 1000 | 7.5 |
| $k = 12$ | 1024 | $5 \times 10^{-4}$ | 1000 | 7.9 |
| $H = 2$ | 1024 | $5 \times 10^{-4}$ | 1000 | 9.9 |
| $H = 3$ | 1024 | $5 \times 10^{-4}$ | 1000 | 10.3 |
| $H = 4$ | 1024 | $5 \times 10^{-4}$ | 1000 | 10.7 |
| $H = 2, I = 1$ | 1024 | $5 \times 10^{-4}$ | 1000 | 13.1 |
| $H = 3, I = 1$ | 1024 | $5 \times 10^{-4}$ | 1000 | 11.0 |
| $H = 4, I = 1$ | 1024 | $5 \times 10^{-4}$ | 1000 | 11.3 |
| $H = 4, I = 2$ | 1024 | $5 \times 10^{-4}$ | 1000 | 12.7 |
| baseline $k = 6$ | 256 | $5 \times 10^{-4}$ | 1000 | 2.6 |
| baseline | 1024 | $5 \times 10^{-4}$ | 1000 | 2.5 |

Table 9: Training details LJ55

| | batch size | learning rate | epochs | training time (h) |
|---|---|---|---|---|
| $k = 7$ | 256 | ($5 \times 10^{-4}$ to $5 \times 10^{-5}$, $5 \times 10^{-5}$) | (150, 850) | 10.5 |
| $k = 27$ | 70 | ($5 \times 10^{-4}$ to $5 \times 10^{-5}$, $5 \times 10^{-5}$) | (150, 69) | 21.9 |
| $k = 55$ | 30 | $5 \times 10^{-3}$ to $1.7 \times 10^{-3}$ | 110 | 32.0 |
| $H = 2$ | 128 | ($5 \times 10^{-3}$ to $5 \times 10^{-4}$, $5 \times 10^{-4}$) | (150, 850) | 16.7 |
| $H = 4$ | 128 | ($5 \times 10^{-3}$ to $5 \times 10^{-4}$, $5 \times 10^{-4}$) | (150, 850) | 16.8 |
| $H = 6$ | 80 | ($5 \times 10^{-3}$ to $5 \times 10^{-4}$, $5 \times 10^{-4}$) | (150, 260) | 11.8 |
| $H = 8$ | 128 | ($5 \times 10^{-3}$ to $5 \times 10^{-4}$, $5 \times 10^{-4}$) | (150, 850) | 14.0 |
| $H = 10$ | 128 | ($5 \times 10^{-3}$ to $5 \times 10^{-4}$, $5 \times 10^{-4}$) | (150, 850) | 14.0 |
| baseline $k = 27$ | 1024 | ($5 \times 10^{-4}$ to $5 \times 10^{-5}$, $5 \times 10^{-5}$) | (150, 850) | 6.6 |
| baseline | 256 | ($5 \times 10^{-4}$ to $5 \times 10^{-5}$, $5 \times 10^{-5}$) | (150, 844) | 24.0 |

All inference details for LJ13, LJ55 and Alanine Dipeptide are reported in tables 11 to 13. The integration of the ODE (eq. (3)) was performed using a fourth order Runge Kutta solver with a fixed step size. We used 20 integration steps.

Table 10: Training details Alanine Dipeptide

| | batch size | learning rate | epochs | training time (h) |
|---|---|---|---|---|
| $H = 2$ | 100 | $(5 \times 10^{-3}$ to $5 \times 10^{-4}, 5 \times 10^{-4})$ | $(150, 850)$ | 20.3 |
| $H = 3$ | 100 | $(5 \times 10^{-3}$ to $5 \times 10^{-4}, 5 \times 10^{-4})$ | $(150, 850)$ | 17.2 |
| $H = 4$ | 128 | $(5 \times 10^{-3}$ to $5 \times 10^{-4}, 5 \times 10^{-4})$ | $(150, 850)$ | 16.7 |
| $H = 5$ | 128 | $(5 \times 10^{-3}$ to $5 \times 10^{-4}, 5 \times 10^{-4})$ | $(150, 850)$ | 16.8 |
| $H = 3, I = 1$ | 80 | $(5 \times 10^{-3}$ to $5 \times 10^{-4}, 5 \times 10^{-4})$ | $(150, 260)$ | 11.8 |
| $k = 11$ | 128 | $(5 \times 10^{-3}$ to $5 \times 10^{-4}, 5 \times 10^{-4})$ | $(150, 850)$ | 14 |
| baseline | 1024 | $5 \times 10^{-4}$ | 1000 | 6.6 |

Table 11: Sampling details LJ13

| | RT (s) | RT$^f$ (s) | RT$^\nabla$ (s) | batch size | GPU usage (%) | GPU mem. usage (%) | # samples |
|---|---|---|---|---|---|---|---|
| $k = 2$ | 407 | 367 | 29 | 40000 | 71 | 66 | $2 \times 10^5$ |
| $k = 4$ | 674 | 577 | 83 | 1024 | 61 | 92 | $2 \times 10^5$ |
| $k = 6$ | 887 | 790 | 71 | 3500 | 73 | 67 | $2 \times 10^5$ |
| $k = 8$ | 1141 | 1023 | 86 | 3500 | 82 | 46 | $2 \times 10^5$ |
| $k = 10$ | 1411 | 1275 | 97 | 3500 | 82 | 86 | $2 \times 10^5$ |
| $k = 12$ | 1532 | 1389 | 103 | 3500 | 83 | 84 | $2 \times 10^5$ |
| $H = 2$ | 1723 | 1527 | 159 | 1024 | 66 | 73 | $2 \times 10^5$ |
| $H = 3$ | 1886 | 1682 | 165 | 1024 | 57 | 42 | $2 \times 10^5$ |
| $H = 4$ | 2050 | 1833 | 174 | 1024 | 54 | 26 | $2 \times 10^5$ |
| $H = 2, I = 1$ | 3581 | 3261 | 242 | 1024 | 79 | 64 | $2 \times 10^5$ |
| $H = 3, I = 1$ | 2488 | 2221 | 207 | 1024 | 66 | 96 | $2 \times 10^5$ |
| $H = 4, I = 1$ | 2395 | 2135 | 202 | 1024 | 60 | 53 | $2 \times 10^5$ |
| $H = 4, I = 2$ | 3197 | 2856 | 257 | 1024 | 59 | 52 | $2 \times 10^5$ |
| baseline $k = 6$ | 4599 | 36 | 4450 | 12000 | 98 | 64 | $2 \times 10^5$ |
| baseline | 2508 | 29 | 2425 | 1000 | 85 | 18 | $1 \times 10^5$ |

Table 12: Sampling details LJ55

| | RT (s) | RT$^f$ (s) | RT$^\nabla$ (s) | batch size | GPU usage (%) | GPU mem. usage (%) | # samples |
|---|---|---|---|---|---|---|---|
| $k = 7$ | 1370 | 1147 | 171 | 256 | 73 | 56 | $4 \times 10^4$ |
| $k = 27$ | 12609 | 11820 | 599 | 70 | 96 | 61 | $4 \times 10^4$ |
| $k = 55$ | 31439 | 30028 | 1114 | 30 | 96 | 83 | $4 \times 10^4$ |
| $H = 2$ | 22602 | 20239 | 2173 | 10 | 79 | 67 | $4 \times 10^4$ |
| $H = 4$ | 20048 | 17897 | 1902 | 15 | 87 | 87 | $4 \times 10^4$ |
| $H = 6$ | 16586 | 14618 | 1707 | 18 | 87 | 84 | $4 \times 10^4$ |
| $H = 8$ | 14960 | 13097 | 1596 | 20 | 87 | 69 | $4 \times 10^4$ |
| $H = 10$ | 13725 | 11982 | 1445 | 25 | 90 | 80 | $4 \times 10^4$ |
| baseline $k = 27$ | 353693 | 348 | 351221 | 256 | 99 | 65 | $4 \times 10^4$ |
| baseline | 529793 | 596 | 525997 | 250 | 99 | 82 | $4 \times 10^4$ |

## D.5 Errors

All errors were obtained as a $68\%$ symmetric percentile around the median from bootstrap sampling using 1000 resampling steps. The reported value is the mean of these bootstrap samples.

Table 13: Sampling details Alanine Dipeptide

| | RT (s) | $RT^f$ (s) | $RT^\nabla$ (s) | batch size | GPU usage (%) | GPU mem. usage (%) | # samples |
|---|---|---|---|---|---|---|---|
| $H = 2$ | 2791 | 2382 | 370 | 100 | 70 | 79 | $5 \times 10^4$ |
| $H = 3$ | 2532 | 2123 | 369 | 100 | 75 | 81 | $5 \times 10^4$ |
| $H = 4$ | 2167 | 1837 | 284 | 128 | 72 | 92 | $5 \times 10^4$ |
| $H = 5$ | 2182 | 1848 | 285 | 128 | 74 | 85 | $5 \times 10^4$ |
| $H = 3, I = 1$ | 3734 | 3236 | 431 | 80 | 79 | 91 | $5 \times 10^4$ |
| $k = 11$ | 1838 | 1633 | 178 | 128 | 78 | 68 | $5 \times 10^4$ |
| baseline | 26554 | 71 | 26094 | 1024 | 99 | 60 | $5 \times 10^4$ |

## D.6 Runtimes

The runtimes were computed by directly measuring the wall-clock time of the computation of interest. Due to missing data, the runtimes of the forward and backward pass in fig. 3(b) needed to be extrapolated from a different run with the same model by assuming the same ratios to the known total runtime.

## D.7 Weight Clipping

As described in section 5, we remove a left and right percentile (i.e., one percent on each side) of the log importance weights, $\log w_i$, to make the ESS and EffSU estimations more robust. The corresponding quantities are named $ESS_{rem}$ and $EffSU_{rem}$. While these metrics are biased, this procedure was necessary to obtain a reliable and robust estimate of the effective speed-up of hollow flow as reflected in the errors of the right most column of tables 4 to 6.

## D.8 Computing Infrastructure

All experiments were conducted on GPUs. The training and inference for LJ13 and Alanine Dipeptide was conducted on a *NVIDIA Tesla V100 SXM2* with 32GB RAM. The training for LJ55 was conducted on a *NVIDIA Tesla A100 HGX* with 40GB RAM, inference was performed on a *NVIDIA Tesla A40* with 48GB RAM.

