# OpenReview forum: "HollowFlow: Efficient Sample Likelihood Evaluation using Hollow Message Passing"
_NeurIPS.cc/2025/Conference — NeurIPS 2025 poster_

### Official Review · Reviewer_KaKL · 2025-06-27

**Clarity:** 3
**Significance:** 2
**Originality:** 4
**Rating:** 5
**Confidence:** 3

**Summary:**

This paper addresses the problem of expensive divergence computations for likelihood evaluations in continuous normalizing flows. The solution adapts a previously proposed network architecture which can report its Jacobian trace in a single backward pass. The present authors describe the construction of a similar architecture with equivariant GNNs. The resulting network is less powerful than standard GNNs, but the fast likelihood computation enables fast ESS/s on the well-studied LJ-13 LJ-55 systems.

**Questions:**

See above.

**Ethical Concerns:**

["NO or VERY MINOR ethics concerns only"]

**Final Justification:**

I maintain my positive assessment of the technical aspects of the work from my original review.

**Limitations:**

Yes

**Quality:**

3

**Strengths And Weaknesses:**

**Strengths**
* The paper treats an important and very unappreciated problem in the Boltzmann sampling literature, which is the difficulty of scaling exact likelihood CNFs to high dimensions despite their superior performance to coupling flows. There are few obvious ways of tackling this challenge, resulting in fewer papers published in this area. The authors' present work is among the more creative recent contributions to Boltzmann sampling.
* Although the basic idea of HollowNet is pre-existing, the authors present a highly nontrivial adaptation to GNNs operating equivariantly via careful construction of the message passing graph. The runtime characteristics of this network are carefully considered and formalized.
* The empirical results indeed show that trading off accuracy for fast likelihoods is sometimes worthwhile, supporting the promise of the research direction.

**Weaknesses**
* It is not acceptable to manipulate the ESS metrics by excluding outliers, a protocol which has no justification in importance sampling. **The authors must correct this, or I will lower the score.**
* The performance penalty is larger than one would hope, calling into question whether the method would work for more difficult systems.
* The authors should discuss considerations, if any, in adapting the framework to non-equivariant GNNs or transformers, as these are increasingly the architectures of choice for molecular diffusion or flows.
* The kNN graph construction is non-differentiable. Can the authors comment on how this changes the validity of the divergence computation or the CNF?

---

> ### Author Rebuttal · Authors · 2025-07-31
>
> We thank the reviewer for the valuable, detailed feedback and thoughtful questions. Answers to the reviewer's questions and additional comments can be found below.
>
> 1.
> >It is not acceptable to manipulate the ESS metrics by excluding outliers, a protocol which has no justification in importance sampling. The authors must correct this, or I will lower the score.
>
> We agree that 'manipulating' benchmark statistics is not acceptable. We however emphasize that we include these to illustrate the impact of the outliers on the ESS, and how controlling these is a path forward for improving the presented approach
>
> 2.
> > The performance penalty is larger than one would hope, calling into question whether the method would work for more difficult systems.
>
> We agree that the penalty is large, however, we expect this to be linked to the locality assumption which we use throughout the manuscript. To test this, we will include an additional ablation study in the camera ready version where we use a KNN graph for the baseline to disentangle the contributions of locality and HoMP on the performance penalty.
>
> Additionally, for the camera ready version, we will implement multi-headed HoMP similar to the multi-head attention described in appendix A , where several relatively sparse HoMP with different connectivities (e.g. separated length scales) are run in parallel and combined in the end. This makes sure that each network does not get disconnected as fast while not having to do any locality assumption. Furthermore, we are currently trying HollowFlow on Alanine Dipeptide and will include the results in the camera ready version.
>
> 3.
> >The authors should discuss considerations, if any, in adapting the framework to non-equivariant GNNs or transformers, as these are increasingly the architectures of choice for molecular diffusion or flows.
>
> By non-equivariant, we assume the reviewer means, not equivariant with respect to the euclidean groups, e.g. E(3), as explored in our examples. This can be done by simply adopting a non-equivariant MP/transformer architecture as is, but ensuring the MP/attention is done in a manner that is faithful to the HoMP recipe, as outlined in the paper. Relaxing permutation symmetry can be done by avoiding using the same embeddings for the same particles/nodes in the graph.
>
> 4.
> >The kNN graph construction is non-differentiable. Can the authors comment on how this changes the validity of the divergence computation or the CNF?
>
> It is unclear what the reviewer means here. The KNN graph is used to limit the connectivity of the MP graph, and the divergence calculation does not depend on the graph construction once it is set.

---

> > ### Comment · Reviewer_KaKL · 2025-08-06
> >
> > I appreciate the authors' response, but they do not sufficiently address my concerns.
> >
> > 1. As originally mentioned, the manipulation of ESS metrics is inappropriate and I do not find the authors' defense convincing. I would request that the authors present new results with the unadjusted ESS metrics. If this is not done, I will unfortunately have to recommend against acceptance. I like the idea very much and respect the technical depth of the results, but transparent and honest reporting of results is paramount.
> >
> > 4. The divergence is defined only for a differentiable vector field. If the vector field is constructed via a GNN over a kNN graph, the neural network output is not continuous wrt to the inputs. Can the authors comment on how this affects the validity of using the divergence to compute model likelihoods?

---

> > > ### Author Response · Authors · 2025-08-07
> > >
> > > 1. We want to emphasize that we do report the original ESS without any outlier removal for all the experiments in table 2 and table 3. We do, however, acknowledge that the comparison between the baseline and HollowFlow is done using the $ESS_{rem}$, i.e. the ESS computed with a percentile of outliers removed. For completeness, we provide additional results for LJ13 below, where we recompute the speed-up that HollowFlow provides using the original ESS. The numbers in brackets indicate a 68% confidence interval calculated via bootstrapping. $ESS_{rel}$ is defined analogously to $ESS_{relrem}$ (eq. 22):
> > >
> > >  $ESS_{rel} = \frac{ESS  \sharp samples}{RT\cdot{(GPU usage)}}$
> > >
> > > Note that some of the numbers differ from the original paper as we retrained the baseline model using best model selection techniques. As removing outliers improves the baseline model more than it does improve the HollowFlow models, the effective speed-up with outliers is even larger than the speed-up without outliers. We will include these results, as well as the corresponding results for LJ55, in the camera ready version and clarify the context. We will also rerun the calculations with more samples to decrease errors.
> > >
> > > |      | ESS                  | ESS$_{rem} \ (\uparrow)$ | ESS$_{rel} \ (\uparrow)$   | ESS$_{relrem} \ (\uparrow)$ | Effective speed-up without outliers$ \ (\uparrow)$ | Effective speed-up with outliers$ \ (\uparrow)$ |
> > > |----------|----------------------|----------------------------|------------------------------|-------------------------------|------------------------------------------------------|---------------------------------------------------|
> > > | k=2      | $0.05\;(0.04, 0.06)$ | $2.92\;(2.89, 2.96)$       | $36.88\;(27.30, 46.49)$      | $2022.97\;(2002.52, 2041.06)$ | $1.059\;(1.048, 1.070)$                              | $1.17\;(0.32, 1.93)$                              |
> > > | k=4      | $0.04\;(0.00, 0.10)$ | $10.35\;(10.29, 10.41)$    | $20.93\;(0.43, 49.26)$       | $5063.54\;(5035.91, 5091.21)$ | $2.651\;(2.633, 2.667)$                              | $0.57\;(0.01, 1.36)$                              |
> > > | k=6      | $3.24\;(2.35, 4.22)$ | $20.20\;(20.11, 20.28)$    | $1002.61\;(735.85, 1315.62)$ | $6229.21\;(6205.99, 6251.38)$ | $3.261\;(3.245, 3.280)$                              | $31.04\;(10.10, 47.89)$                           |
> > > | k=8      | $2.63\;(2.22, 3.01)$ | $20.63\;(20.56, 20.72)$    | $574.24\;(485.64, 658.83)$   | $4415.12\;(4396.01, 4431.49)$ | $2.311\;(2.299, 2.323)$                              | $18.26\;(6.13, 29.14)$                            |
> > > | k=10     | $1.06\;(0.77, 1.37)$ | $16.50\;(16.44, 16.57)$    | $185.75\;(136.43, 231.83)$   | $2834.40\;(2820.80, 2848.38)$ | $1.484\;(1.476, 1.493)$                              | $6.01\;(1.84, 10.44)$                             |
> > > | k=12     | $4.21\;(3.38, 4.95)$ | $19.73\;(19.65, 19.79)$    | $639.88\;(484.01, 786.38)$   | $3105.94\;(3095.25, 3119.05)$ | $1.626\;(1.617, 1.634)$                              | $20.72\;(6.58, 33.67)$                            |
> > > | baseline | $1.46\;(0.42, 2.02)$ | $40.75\;(40.64, 40.87)$    | $85.48\;(19.22, 94.60)$      | $1910.42\;(1903.62, 1916.60)$ | $1$                              | $1$                              |
> > >
> > >
> > >
> > > 2. We thank the reviewer for clarifying the question. We agree that if the $k$NN graph is updated dynamically, non-differentiability occurs on a set of measure-zero hypersurfaces (the decision boundaries of the $k$NN construction). This can cause theoretical issues, including but not limited to the uniqueness and existence of a global solution to the corresponding ODE. However, empirically, we observe no instability. We expect this to be the case as the vector field remains differentiable almost everywhere, and consequently the ODE and the corresponding continuity equation is well-defined almost everywhere. We acknowledge that this is a somewhat heuristic explanation, but defer formal analysis of these cases, to explain our empirical results, to future work, as the specific graph construction is not a central contribution to this work.
> > >     To further validate empirically that the non-differentiability does not cause any problems in practice, we tracked the divergence of a trained fully connected and a trained 6 nearest neighbors ($6$NN) HollowFlow model during sampling using a forward Euler scheme with 20 integration steps. We monitor the maximal and average absolute difference of the divergence of consecutive steps over 1000 initial conditions to measure if there are significant discontinuities in the $6$NN model as opposed to the fully connected one. Results can be found below. While both values are slightly higher for the $6$NN model, there are no significant differences further supporting our claim that the non-differentiability does not cause issues in practice.
> > >     |  | max difference | average difference |
> > >     |---|---|---|
> > >     | fully connected | 72.5 | 5.3 |
> > >     | k = 6 | 74.2 | 5.8 |

---

### Official Review · Reviewer_tKoR · 2025-06-30

**Clarity:** 3
**Significance:** 3
**Originality:** 3
**Rating:** 4
**Confidence:** 2

**Summary:**

This paper introduces "HollowFlow," a novel continuous normalizing flow (CNF) architecture designed to drastically reduce the computational cost of likelihood evaluation for generative models, particularly in the context of Boltzmann Generators (BGs). The authors' key contribution is a new message passing scheme called Hollow Message Passing (HoMP) addressing the prohibitive scaling of computing the divergence of the vector field in CNFs, which traditionally requires $O(N)$ backward passes.

HoMP is implemented using a non-backtracking graph neural network (NoBGNN) that operates on a line graph representation of the system's connectivity. This construction splits the Jacobian of the neural network vector field into a block-diagonal part and a "block-hollow" part, which allows for the exact computation of the Jacobian's trace (the divergence) with a constant number (dimension of coordinates) of backward passes.

The authors provide a theoretical analysis of the computational complexity, showing that HollowFlow can achieve a speed-up of up to $O(n^2)$ over standard equivariant GNN-based flows. They validate their approach by training E(3)-equivariant HollowFlow models on two Lennard-Jones (LJ) particle systems, LJ13 and LJ55.

**Questions:**

- The appendix clarifies the update rule for the backtracking array B(t), which is of size $n^3$. Could you comment on the practical computational and memory cost of storing and updating this array in your implementation? Could it become a practical bottleneck for systems with thousands of nodes, and do you see a path to optimizing it?
- Your results show a non-monotonic dependence of model performance on the hyperparameter 'k'.  Do you have any intuition or a proposed heuristic for selecting an optimal 'k' for a new system without having to train multiple models? Is there a risk that for a complex landscape, the optimal 'k' would need to be so large that it negates the computational advantage of using a sparse graph?
- Could you elaborate on the training of the baseline model? Given its very low ESS, do you believe it was fully converged? How does its ESS compare to other published flow-based Boltzmann Generators for these LJ systems? A stronger baseline would make the "effective speed-up" claims even more compelling.
- This work is motivated by achieving exact divergence calculation efficiently. How do you see your method comparing to those that use stochastic trace estimators (e.g., Hutchinson's estimator)? While approximate, they are also computationally cheap. In your opinion, what are the key applications or systems where the exactness of the divergence provided by HollowFlow is critical and approximation is insufficient?

**Ethical Concerns:**

["NO or VERY MINOR ethics concerns only"]

**Final Justification:**

The authors address most of my concerns. However, I'm still concerned about the scalability of the method.

**Limitations:**

The authors provide an honest discussion of the limitations in the main paper, which I find accurate. Most of my main concerns align with theirs, and are:
- The current kNN-based implementation assumes locality and is not well-suited for systems with critical long-range interactions (e.g., electrostatics).  This significantly limits its immediate applicability to a wide range of important molecular systems.
- The HoMP algorithm relies on updating a backtracking array, B(t), to dynamically remove edges from the line graph at each message passing step.  This array is of size $O(n^3)$. This could become a new computational or memory bottleneck for very large systems (e.g., n > 1000), potentially undermining the overall speed-up. The appendix provides the algorithm but does not analyze the practical cost of this step.
- The core idea of HoMP is to restrict information flow to gain computational efficiency.  This is an explicit trade-off. It remains an open question whether this restriction will prevent the model from learning the complex energy landscapes of more challenging systems, leading to a sample quality (ESS) so poor that even a massive speed-up cannot compensate for it.

**Quality:**

2

**Strengths And Weaknesses:**

### Strengths:
- The proposed HoMP algorithm is a clever adaptation of the "hollow network" concept to the crucial domain of equivariant GNNs. This offers a direct and effective solution to the divergence calculation bottleneck in CNFs.
- The paper is well-supported by theory and offers step-by-step proofs in the appendix.
- The experimental results are impressive, demonstrating a speed-up of nearly 90x for the LJ55 system.
- The paper provides code and all training details for reproducibility.\
- The paper provides an extension to attention architectures offering integration with various SOTA architectures.

### Weaknesses:
- A notable weakness is the performance of the baseline model, particularly in terms of the effective sample size (ESS). For LJ13, the baseline achieves an ESS of only 0.001%, and for LJ55, it's 0.167%.  These are very low values, suggesting the baseline model is not particularly strong. While the authors acknowledge this, it tempers the impact of the "effective speed-up" metric, as the comparison would be more compelling against a state-of-the-art baseline with a higher ESS.
- The experiments are confined to two relatively small, well-behaved systems (LJ13 and LJ55) that are dominated by short-range interactions. The paper's claims of general applicability to "high-dimensional scientific problems" would be much stronger if tested on more diverse and challenging systems.
- The use of a kNN graph is a necessary compromise to avoid the poor scaling of the line graph construction on a fully connected graph.  However, the choice of 'k' is a critical, non-trivial hyperparameter. Figure 3(c) shows that performance is quite sensitive to 'k', and the appendix does not offer a heuristic for selecting it, adding a layer of tuning complexity.

---

> ### Author Rebuttal · Authors · 2025-07-31
>
> We thank the reviewer for taking the time to give valuable and very detailed feedback. Answers to the reviewer's questions can be found below. Finally, we additionally comment on some limitations that the reviewer pointed out.
>
> Questions
>
> 1.
> >The appendix clarifies the update rule for the backtracking array B(t), which is of size $n^3$. Could you comment on the practical computational and memory cost of storing and updating this array in your implementation? Could it become a practical bottleneck for systems with thousands of nodes, and do you see a path to optimizing it?
>
> In practice, it is sufficient to only save information about from which nodes of $G$ all the nodes of $LG$ have received information from. If $G$ has $n$ nodes and $LG$ has $n_{lg}$ nodes, the size of the array will effectively be $nn_{lg}$. For a KNN graph, $n_{lg} = \mathcal{O}(nk)$ so the memory requirements of $B(t)$ in our implementation scale as $n^2k$. In practice, only indexing operations, element wise binary logical functions and PyTorch scatter-add operations are applied to $B(t)$. Those do, to the best of our knowledge, not scale more than linearly with the size of $B(t)$ and can be (and probably are internally in PyTorch) heavily parallelized. We never experienced any computational bottlenecks during these calculations.
> If it does become a problem, one possible strategy to circumvent it is to use a graph $G$, that has a fixed girth (length of the shortest cycle) of at least $T^{lg} + 2$, where $T^{lg}$ is the number of message passing (MP) steps on the line graph. This ensures that the Jacobian stays hollow even without removing connections rendering $B(t)$ unnecessary.
>
> 2.
> >Your results show a non-monotonic dependence of model performance on the hyperparameter 'k'. Do you have any intuition or a proposed heuristic for selecting an optimal 'k' for a new system without having to train multiple models? Is there a risk that for a complex landscape, the optimal 'k' would need to be so large that it negates the computational advantage of using a sparse graph?
>
> Naively, one might expect that the model performance (ESS) would monotonically increase with $k$ as there are more connections in the graph for larger $k$. However, this is not the case in practice (see e.g. table 3). The fact that the ESS does not increase further with $k$ beyond some upper bound can heuristically be explained as follows: By construction, some of the connections in the line graph need to be removed after a certain number of MP steps (See appendix B for more details). The larger $k$, the more connections need to be removed after only one MP step up to the point where (almost) no connections are left after only one MP step. We thus have two counteracting effects when $k$ increases: The first MP step gains more connections and thus expressiveness while the second MP step looses connections and expressiveness. Eventually, the latter seems to take over effectively limiting the expressiveness of the model.  We will include a brief summary of this explanation in the camera ready version to improve clarity.
>
> As larger $k$'s are not necessarily better, we do not think that for complex landscapes the optimal $k$ will be so large that it negates the computational advantage.
>    To (heuristically) choose $k$, one can efficiently numerically work out how many edges of the line graph are left after $T$ message passing steps without training a model. There should be a significant amount of edges left after one MP step, and the graph should not be extremely sparse in the beginning. We will include a plot showing the percentage of edges remaining as a function of the number of message passing steps for different $k$'s together with an instruction on how to tune $k$ in the appendix of the camera ready version.
>
> 3.
> >Could you elaborate on the training of the baseline model? Given its very low ESS, do you believe it was fully converged? How does its ESS compare to other published flow-based Boltzmann Generators for these LJ systems? A stronger baseline would make the "effective speed-up" claims even more compelling.
>
> The baseline model was trained in a similar way as in the Equivariant Flow Matching paper (Klein et al., 2023). However, there are some significant differences. Beside a different learning rate scheduler (see appendix C4), most importantly, we use a different architecture (PaiNN) instead of their architecture that we will refer to as EGNN. The ESS for e.g. LJ13 that was reported in (Equivariant Flow Mathching, Klein et al., 2023) is almost 58\% while our ESS is close to zero if we do not remove a small number of outliers and comparable (49\%) if we do remove these outliers. As the two architectures are different (different update rules, different embeddings etc.), there might be non-trivial effects that could explain the difference in ESS, similar to the non-linear scaling with $k$.
>
>    While we do believe that our baseline models are fully converged, it might be possible to further increase performance with different training strategies and choice of hyperparameters.
>
>    It is important to keep in mind that the ESS is extremely sensitive to outliers of the log importance weight histogram. To illustrate this effect, we included $ESS_{rem}$ where these outliers are removed in a controlled way.
> Future work will be necessary to fully understand how different architectures used in HollowFlow and in the non-hollow baseline do affect the ESS and especially how different architectures influence the amount of outliers that significantly lower the ESS.
>
> 4.
> > This work is motivated by achieving exact divergence calculation efficiently. How do you see your method comparing to those that use stochastic trace estimators (e.g., Hutchinson's estimator)? While approximate, they are also computationally cheap. In your opinion, what are the key applications or systems where the exactness of the divergence provided by HollowFlow is critical and approximation is insufficient?
>
> For Boltzmann generator applications, variance in the likelihood estimator leads to biased weights under self-normalized importance sampling. Thus, exact likelihoods and exact divergences are critical and even an unbiased estimator like the Hutchinson estimator is insufficient. Furthermore, considering how sensitive the ESS is to even a small number of outliers, it is important to get the importance weights exactly right.
>
>
> Limitations:
>
> >The current kNN-based implementation assumes locality and is not well-suited for systems with critical long-range interactions (e.g., electrostatics). This significantly limits its immediate applicability to a wide range of important molecular systems.
>
> While we agree with the reviewer's observation that we currently assume locality, we want to emphasize that this can be circumvented by a multi-head strategy, where several relatively sparse HoMP with different connectivities (e.g. separated length scales) are run in parallel and combined in the end (see appendix A for a brief discussion). This makes sure that each network does not get disconnected as fast while not having to do any locality assumption. This introduces an additional scaling with the number of heads while the scaling with the number of particles remains the same for each head. We will include a more detailed discussion of multi-head HollowFlow in the camera ready version.
>
> >The core idea of HoMP is to restrict information flow to gain computational efficiency. This is an explicit trade-off. It remains an open question whether this restriction will prevent the model from learning the complex energy landscapes of more challenging systems, leading to a sample quality (ESS) so poor that even a massive speed-up cannot compensate for it.
>
> We are currently working on running HollowFlow on Alanine Dipeptide and plan to include the results in the camera ready version.

---

> > ### Comment · Reviewer_tKoR · 2025-08-02
> >
> > I thank the authors for their detailed rebuttal. This clears up most of my concerns and I'd be happy to increase my score.

---

> > > ### Author Response · Authors · 2025-08-03
> > >
> > > We thank the reviewer for engaging in the discussion phase and for increasing their score. We remain open to discussion in case any doubts remain.

---

### Official Review · Reviewer_PymQ · 2025-07-02

**Clarity:** 2
**Significance:** 2
**Originality:** 3
**Rating:** 4
**Confidence:** 3

**Summary:**

This paper proposes HollowFlow, a framework that combines an non-backtracking GNNs and some NNs to speedup sampling and likelihood evaluation in previous BG methods. Specifically, HollowFlow's non-backtracking GNNs enforce block-diagonal Jocabian structure, which effectively reduces the number of backward pass. Experiments conducted on LJ13 and LJ55 further validate the effectiveness of HollowFlow.

**Questions:**

See above

**Ethical Concerns:**

["NO or VERY MINOR ethics concerns only"]

**Final Justification:**

most concerns have been solved. i raise the score.

**Limitations:**

See above

**Quality:**

3

**Strengths And Weaknesses:**

Strengths:
1. HollowFlow innovatively introduces HoMP to effectively reduces the number of backward passes required for the likelihood evaluations. KNN graphs are also incorporated to reduce the graph complexity.

2. The paper develops theoretical scaling laws that aligns well with experimental results.


Weaknesses:
1. Although the paper states that:

> the performance of our baseline is not in line with state-of-the-art in terms of ESS, however, since our HollowFlows are built directly upon this baseline, we anticipate that any improvement in the baseline will be reflected in the HollowFlow models as well.

Nevertheless, the lack of direct comparison with state-of-the-art methods reduces the persuasiveness of the results regarding HollowFlow's performance. I suggest that the authors supplement their experiments with additional benchmarks against leading approaches.

2. While HollowFlow achieves remarkable speed-ups compared to the baseline, the ESS and ESS_rem is noticably lower. Furthermore, increasing the value of $k$ does not seem to improve HollowFlow's ESS and ESS_rem beyond a certain upper bound, and this upper bound remains significantly lower than that of the baseline. What is the rationale of this phenomenon? and How to tune HollowFlow in practice?

3. The introduction of KNN assumes the locality of the graph, as stated in the limitation section. How does HollowFlow perform with a fully connected graph? Additionally, I recommend including an ablation study to disentangle the contributions of HoMP, KNN, etc.

4. Experiments are conducted solely on LJ systems. Thus, the generalizability of the proposed model seems not sufficiently validated.

---

> ### Author Rebuttal · Authors · 2025-07-31
>
> We thank the reviewer for the valuable and very detailed feedback. Answers to the reviewer's questions and comments can be found below.
>
> 1. Thanks for your suggestion, a direct comparison against a state-of-the-art method might indeed be interesting. However, we want to emphasize that our goal is not to beat the state of the art in terms of ESS but to demonstrate a significant improvement in terms of ESS per compute during sampling for a given architecture. In our experiments, we decided to use the PaiNN architecture as a base to implement HoMP even though state-of-the-art results have been achieved with other architectures like the architecture, that we will refer to as EGNN in the following, used in the Equivariant Flow Matching paper (Klein et al., 2023). HoMP can also be implemented with EGNN but in contrast to PaiNN it requires some arbitrary symmetry breaking on the line graph as all terms in the message passing will cancel out otherwise. While a base line using EGNN achieves an ESS that is close to the state-of-the-art, it seems to lower performance of HollowFlow, presumably due to the aforementioned symmetry breaking. Thus, we stuck with PaiNN for our baseline and HollowFlow, even though the PaiNN baseline is not in line with state-of-the-art methods.
>
>
> 2. HollowFlow uses a restricted architecture that we expect to have a limited expressiveness resulting in a lower ESS. However, as pointed out above, our goal is to improve the ESS per compute during sampling which we still achieve due to enormous computational savings. The fact that the ESS does not increase further with $k$ beyond some upper bound can heuristically be explained as follows: By construction, some of the connections in the line graph need to be removed after a certain number of message passing (MP) steps (See appendix B for more details). The larger $k$, the more connections need to be removed after only one MP step up to the point where (almost) no connections are left after only one MP step. We thus have two counteracting effects when $k$ increases: The first MP step gains more connections and thus expressiveness while the second MP step looses connections and expressiveness. Eventually, the latter seems to take over effectively limiting the expressiveness of the model.  We will include a brief summary of this explanation in the camera ready version to improve clarity.
>
>    As stated in table 3, we found empirically that HoMP with relatively small $k$'s ($k=6, k=8$ for LJ13) seems to work about as well as fully connected HoMP ($k=12$). To (heuristically) tune $k$ in practice, one can efficiently numerically work out how many edges of the line graph are left after $T$ message passing steps without training a model. There should be a significant amount of edges left after one MP step and the graph should not be extremely sparse in the beginning. We will include a plot showing the percentage of edges remaining as a function of the number of message passing steps for different $k$’s in the appendix of the camera ready version together with an instruction on how to tune $k$.
>
>    In practice, one could also use a multi-head strategy where several relatively sparse HoMP with different connectivities (e.g. separated length scales) are run in parallel and combined in the end (see appendix A for a brief discussion). This makes sure that each network does not get disconnected as fast while not having to do any locality assumption. This introduces an additional scaling with the number of heads while the scaling with the number of particles remains the same for each head. We will include a more detailed discussion of multi-head HollowFlow in the camera ready version.
>
> 3. While we indeed assume locality of the graph, it is possible to circumvent this assumption by the multi-head approach described above. In Table 3 we show how HollowFlow performs with a fully connected graph on LJ13 (see row labelled $k=12$). This can already be seen as an ablation to dissentangel the contribution of the locality assumption. We will add a similar study for LJ55. Running HollowFlow without HoMP is not possible as HoMP is inevitable to achieve the hollow structure of the Jacobian unless we only do one message passing step. We will include an additional ablation study in the camera ready version where we run the baseline with a KNN graph.
>
> 4. It would indeed be desirable to test the method on other systems. We are currently working on running HollowFlow on Alanine Dipeptide and will include the results in the camera ready version.

---

> > ### Comment · Reviewer_PymQ · 2025-08-06
> >
> > thx for the rebuttal. most of my concerns have been solved. I raise my score to 4, good luck

---

> > > ### Author Response · Authors · 2025-08-07
> > > **Thanks for your engagement**
> > >
> > > We thank the reviewer for their engagement and feedback on our paper. If any outstanding questions remain we are happy to engage with them until the end of the rebuttal period.

---

### Note · Authors · 2025-08-12

We thank the reviewers for their engagement and the valuable and detailed feedback. Below, we give a brief summary of the concerns raised and how we address them.

**1.Non-linear dependence of the model on the hyper parameter $k$:**
We gave a detailed explanation of the non-linear behaviour along with an instruction on how to tune $k$.
Camera ready:  We will include tuning instructions, a plot showing how the graph disconnects for different values of $k$ and a detailed explanation.

**2. Locality assumption of the $k$NN graph:**
We pointed out a multi-head strategy, already briefly described in appendix A, to mitigate that.
Camera ready: We will add a discussion and experimental results of the multi-head approach, showing preliminary twofold performance gains on LJ13.

**3. Ablation studies to disentangle $k$NN  and HoMP restrictions:**
We agreed that further ablations are possible and worthwhile.
Camera ready: We will add ablations running the baseline with a $k$NN graph for LJ55 and LJ13 as well as HoMP with a fully connected graph for LJ55.

**4. Performance on more complex systems:**
We expect expressivity issues mainly stem from the $k$NN locality assumption, which the multi-head strategy should alleviate.
Camera ready: We plan to add results for (multi-head) HollowFlow on Alanine Dipeptide.

**5. Cubic memory scaling of non-backtracking array:**
While one reviewer suggested $\mathcal{O}(n^3)$ scaling, our implementation scales as $\mathcal{O}(n^2k)$. In addition, high parallelizability should keep computational costs manageable.

**6. Advantage over unbiased trace estimators (e.g., Hutchinson):**
Exact likelihoods and thus exact divergences as provided by HollowFlow are critical for Boltzmann generator applications.

**7. ESS without outlier removal:**
We removed a small percentile of outliers to stabilize the ESS due to its sensitivity. For completeness, we recomputed speed-up claims without outlier removal.
Camera ready: We will include these results for all experiments.

**8. Adaptation to non-equivariant GNN and transformer architectures:**
This is straightforward and possible.
Camera ready: We will clarify adaptation procedures.

**9. Discontinuity of the vector field due to $k$NN graph:**
Discontinuities occur on measure-zero hyper surfaces and have not caused practical issues. We support this with numerical evidence.
Camera ready: A discussion and numerical observations will be added to the appendix.

---

### Decision · Program_Chairs · 2025-09-17

**Decision:**

Accept (poster)

**Comment:**

The authors propose a flow-based generative model based on a GNN that makes sample likelihood computations more efficient, using a novel non-backtracking graph neural network (NoBGNN) enforcing a block-diagonal Jacobian structure. They showed speedups on two Lennard-Jones particle systems. Reviewers found the problem to be important and the experimental results significant. Most questions/concerns were resolved during discussion, after which all reviewers leaned towards acceptance. One major point of discussion was the reporting of ESS metrics including outliers (KaKL), which I remind the authors to incorporate into the paper.